



# Improving global paleogeography since the late Paleozoic using paleobiology

Wenchao Cao*,[1], Sabin Zahirovic[1], Nicolas Flament[†,1], Simon Williams[1], Jan Golonka[2] and R. Dietmar Müller[1]

[1] EarthByte Group, School of Geosciences, The University of Sydney, NSW 2006, Australia

[2] Faculty of Geology, Geophysics and Environmental Protection, AGH University of Science and Technology, Mickiewicza 30, 30-059 Kraków, Poland

*Correspondence to: Wenchao Cao (wenchao.cao@sydney.edu.au)

[†]Current address: School of Earth and Environmental Sciences, University of Wollongong, Northfields Avenue, Wollongong, New South Wales 2522, Australia

**Abstract.** Paleogeographic reconstructions are important to understand Earth's tectonic evolution, past

eustatic and regional sea level change, hydrocarbon genesis, and to constrain and interpret the dynamic topography predicted by time-dependent global mantle convection models. Several global paleogeographic maps have been compiled and published but they are generally presented as static maps with varying temporal resolution and fixed spatial resolution. Existing global paleogeographic maps are also tied to a particular plate motion model, making it difficult to link them to alternative

digital plate tectonic reconstructions. To address this limitation, we developed a workflow to reverse-engineer global paleogeographic maps to their present-day coordinates and enable them to be linked to any tectonic reconstruction. Published paleogeographic compilations are also tied to fixed input datasets. We used fossil data from the Paleobiology Database to identify inconsistencies between fossils paleo-environments and published paleogeographic maps, and to improve the location of

inferred terrestrial-marine boundaries by resolving these inconsistencies. As a result, the overall consistency ratio between the paleogeography and fossil collections was improved from 76.9% to 96.1%. We estimated the surface areas of global paleogeographic features (shallow marine environments, landmasses, mountains and ice sheets), and reconstructed the global continental flooding history since the late Paleozoic based on the amended paleogeographies. Finally, we discuss the

relationships between emerged land area and total continental crust area through time, continental growth models, and strontium isotope ($^{87}Sr/^{86}Sr$) signatures in ocean water. Our study highlights the flexibility of digital paleogeographic models linked to state-of-the-art plate tectonic reconstructions in order to better understand the interplay of continental growth and eustasy, with wider implications for understanding Earth's paleotopography, ocean circulation, and the role of mantle convection in shaping

long-wavelength topography.

## 1 Introduction

Paleogeography is widely used in a range of fields including paleoclimatology, plate tectonic

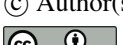



40 reconstructions, paleobiogeography, resource exploration and geodynamics. Several global deep-time

paleogeographic compilations have been published (e.g. Blakey, 2008; Golonka et al., 2006; Ronov, et

al., 1984, 1989; Scotese, 2004; Smith et al., 1994). However, they are generally presented as static

paleogeographic snapshots with varying temporal resolution and fixed spatial resolution, and are tied to

a particular plate motion model. This makes it difficult to link them to alternative digital plate tectonic

45 reconstructions, and to update paleogeographic maps when plate motion models are improved. It is

therefore challenging to use paleogeographic maps to help constrain or interpret numerical models of

mantle convection that predict long-wavelength topography (Gurnis et al., 1998; Spasojevic and Gurnis,

2012) based on different tectonic reconstructions, or as an input to models of past ocean and

atmosphere circulation/climate (Goddéris et al., 2014; Golonka et al., 1994) and models of past

50 erosion/sedimentation (Salles et al., 2017).

In order to address this issue, we developed a workflow to reverse-engineer published

paleogeographies to their corresponding present-day coordinates so that the geometries could be

attached to any plate motion model. This was the first step towards the construction of paleogeographic

55 maps with flexible spatial and temporal resolutions that are more easily testable and expandable with

the incorporation of new paleo-environmental datasets (e.g. Wright et al., 2013). In this study, we used

a set of global paleogeographic maps (Golonka et al., 2006) covering the entire Phanerozoic time

period as the base paleogeographic model. We reverse-engineered these global paleogeographic maps

to their present-day coordinates and then reconstructed them using the plate motion model of Matthews

60 et al. (2016). Subsequently, we used fossil data from the Paleobiology Database (https://paleobiodb.org)

to identify inconsistencies between fossils paleo-environments and the paleogeographic maps, and to

improve the location of inferred terrestrial-marine boundaries by resolving these inconsistencies.

Finally, we used the improved reconstructed paleogeographies to estimate the surface areas of global

paleogeographic features (shallow marine environments, landmasses, mountains and ice sheets), to

65 investigate the global continental flooding history since the Devonian and compare it with global sea

level change over time (Haq et al., 1987; Haq et al., 2008; Müller et al., 2008). In addition, we

discussed the evolution of the modelled emerged land area and total continental area in connection with

continental growth models, the strontium isotope ($^{87}Sr/^{86}Sr$) signature from the proxy records (Flament

et al., 2013; van der Meer et al., 2017), and the assembly and breakup of Pangea.


**2 Data and Paleogeographic Model**

The data used in this study are global paleogeographic maps and paleontological data for the last 402

Myr, which originate from the set of paleo-maps produced by Golonka et al. (2006) and the

75 Paleobiology Database (https://paleobiodb.org), respectively. The global paleogeographic compilation

by Golonka et al. (2006), spanning the entire Phanerozoic, is divided into 32 time-interval maps using

the time scale of Sloss (1988) (Table 1). Each map is a compilation of paleolithofacies and paleo-

environments for each geological time interval. These paleogeographic reconstructions illustrate the



changing configuration of ice sheets, mountains, landmasses, shallow marine environments (inclusive

of shallow seas and continental slopes) and deep oceans during the last 544 million years.

The paleogeographic maps of Golonka et al. (2006) were constructed using a plate tectonic model

available in the Supplement of Golonka (2007), which described the relative motions between plates

and terranes. In this rotation model, paleomagnetic data were used to constrain the paleolatitudinal

positions of continents and rotation of plates, and hot spots, where applicable, were used as reference

points to calculate paleolongtitudes (Golonka, 2007). This rotation model is necessary to accurately

reverse-engineer these paleogeographies (Golonka et al., 2006) to their present-day coordinates so that

they can be attached to any modern plate motion model. The relative plate motions of Golonka (2006,

2007) are similar to those in Scotese (2004).

Here, we use a global plate kinematic model to reconstruct paleogeographies back in time from

present-day locations. The global tectonic reconstruction of Matthews et al. (2016), with continuously

closing plate boundaries from 410-0 Ma, is primarily constructed from a Mesozoic and Cenozoic plate

model (230-0 Ma) (Müller et al., 2016) and a Paleozoic model (410-250 Ma) (Domeier and Torsvik,

2014). This model is a relative plate motion model that is ultimately tied to Earth's spin axis through an

absolute reference frame (Matthews et al., 2016).


The Paleobiology Database (https://paleobiodb.org) is a compilation of global fossil data covering deep

geological time. All fossils in the database are associated with detailed metadata, including the time

range (typically biostratigraphic age), present-day geographic coordinates, host lithology, and paleo-

environment. Figure 1 visualizes global fossil distribution and shows the total numbers of fossil

collections on Earth since the Devonian period. The documented fossils are unevenly distributed both

spatially and temporally, largely due to the differences in fossil preservation and the spatial sampling

biases of fossil localities. For this study, a total of 57,854 fossil collections with temporal and paleo-

environmental assignments from 402 to 2 Ma were downloaded from the database on 7 September

110     2016.

**3 Method**


The methodology mainly involves the processes of paleogeographic reverse-engineering, subsequent

reconstructing in another rotation model and eventually improving using paleobiology data. Figure 2

illustrates a generalized workflow that can be applied to any paleogeography model. In order to



represent the paleogeographic maps as digital geographic geometries, they are first georeferenced using the original projection and coordinate system (such as global Mollweide in Golonka et al., 2006), and then reprojected into the WGS84 geographic coordinate system. The resulting maps are then attached to the original rotation model using the open-source and cross-platform plate reconstruction software, GPlates (www.gplates.org). Every plate is then assigned a unique plate ID that defines the rotation rules in geological times so that the paleogeographies can be rotated back to their present-day coordinates (see example in Figs. 3a, b). We use present-day coastlines and terrane boundaries with the plate IDs of Golonka (2007) as a reference to refine rotations and ensure a high accuracy of the reverse engineering.

When the paleogeographic maps in present-day coordinates are attached to a new reconstruction model, e.g. Matthews et al. (2016) as used in this study, the resulting paleogeographies contain gaps (Fig. 3c, pink) and overlaps between neighboring polygons, when compared to the original reconstruction (Fig. 3a). These gaps and overlaps essentially arise from the differences in the reconstructions described in Matthews et al. (2016) and Golonka et al. (2006). The reconstruction of Golonka et al. (2006) typically has a tighter fit of the major continents within Pangea prior to the supercontinent breakup. In addition, this reconstruction contains a different plate motion history and block boundaries definitions in regions of complex continental deformation, for example along active continental margins (e.g. Himalayas, western North America, Fig. 3c).

The gaps and overlaps cause changes in the total areas of paleogeographies at different time intervals, becoming larger or smaller, when compared with the original paleo-maps (Golonka et al., 2006). The gaps can be fixed by interactively extending the outlines of the polygons in a GIS platform to make the plates connect as in the original paleo-maps (Fig. 3a, c and d). The resulting paleogeographies with fixed gaps (Fig. 3d) change to different extent in total area compared with the original paleogeographies (Golonka et al., 2006). The total areal variations range from the maximum 5.8% to the minimum -2.7%, with an average of -1.4%. To avoid artefacts introduced from overlapping paleogeographies, the drawing order was standardized using the following sequence: ice sheets, mountains, landmasses and finally shallow marine environments (top to bottom layering).

Once the gaps are fixed, the consistency between the reconstructed paleogeography and paleobiology data can be tested. These tests are aimed at identifying inconsistencies between fossil-derived paleo-environments and underlying paleogeographies in order to improve the accuracy of marine-terrestrial boundaries in the paleogeographic maps. Fossil collections belonging to each time interval (Table 1) are first extracted from the dataset downloaded from the Paleobiology Database. Only the fossils with temporal ranges lying entirely within the corresponding time intervals were selected, as opposed to including the fossils that have larger temporal ranges. Fossils with temporal ranges crossing any time-interval boundary are not taken into consideration. As a result, a minimum number of fossil collections



were selected for each time interval. The selected fossil collections were classified into either terrestrial

or marine setting category, according to a lookup table (Table 2). Alternatively, the terrestrial and

marine fossil data could be separately downloaded from the Paleobiology Database. In this process,

each fossil with a specific environment would be automatically oriented into the corresponding

terrestrial or marine groups based on the same classification scheme (Table 2). Fossil collections would

then be extracted in each time interval (Table 1) from terrestrial and marine fossils subgroups,

respectively.

Fossil collections are then attached to the plate motion model of Matthews et al. (2016) so they can be

reconstructed at each time interval. Subsequently, a point in polygon test is used to verify if the

indicative paleo-environment (terrestrial or marine) of fossil collections is consistent with the

underlying paleogeographic features. In this process, polygons are tested in the following sequence: ice

sheets, mountains, landmasses and shallow marine environments. Terrestrial fossil paleo-environments

correspond to landmass, mountain or ice sheet paleogeography. Fossil shallow marine environments

map to marine environments in paleogeography.

Based on the inconsistencies between fossils paleo-environments and underlying paleogeographies, we

can modify the terrestrial-marine boundaries in the paleo-maps. Figures 4 and 5 illustrate how to

modify the marine-terrestrial boundaries in the paleogeographic maps based on the test results.

Modifications are made according to the following rules: (1) Fossil collections from the Paleobiology

Database are presumed to be well-dated, constrained geographically, not reworked and representative

of their broader paleo-environments. Their indicative environments are assumed to be correct. (2) Only

fossils within 100 km of the nearest terrestrial-marine boundary (for instance, d1 ≤ 100 km in Fig. 4b)

are taken into account as valid proxies to improve marine-terrestrial boundaries. (3) The boundaries are

shifted until the fossils environments are consistent with the underlying paleogeography and at the

same time remain within about 20 km distance from the fossils used (Fig. 4c, d2 ≈ 20 km). (4) The

adjacent boundary is accordingly adjusted and smoothed (Fig. 4c and Fig. 5c). (5) Occasionally, some

adjacent fossils near the same boundary may indicate conflicting paleo-environments. In this case, we

treat these adjacent fossils as a cluster, in which the environment represented by over 50% of fossils is

considered to be indicative of the environment of the entire cluster. For example, the fossils in the

black circle in Fig. 5b are regarded as a cluster, in which over 50% of fossils indicate a shallow marine

environment. These rules are designed to maximize the use of paleobiology to improve paleogeography

while attempting to minimize incorrect modifications. We note that in some cases the paleogeography

cannot be fully reconciled with the Paleobiology Database (for example, inconsistent terrestrial fossils

in the black circle in Fig. 5b).



However, in some rare cases, outlier fossils may be a deceptive recorder of paleogeography. For instance, Wichura et al. (2015) discussed the discovery of a beaked whale fossil 740 km inland from the present-day coastline of the Indian Ocean in the East Africa. The authors found evidence to suggest that this whale could have travelled inland from the Indian Ocean along an eastward-directed fluvial (terrestrial) drainage system and was stranded there, rather than representing a marine setting that

would be implied under our assumptions. Therefore, theoretically, when using paleobiology to improve paleogeography, additional concerns about living habits of fossils and associated geological settings should be taken into account. In this study, we have removed this misleading fossil whale from the dataset. Such instances of deceptive fossils are rare.

**4 Results**

**4.1 Paleobiology Tests**

Global reconstructed paleogeographic maps from 402 to 2 Ma are tested against marine and terrestrial fossils that are reconstructed in the same rotation model (Matthews et al., 2016). The marine fossils

consistency ratio is defined by the marine fossils within shallow marine paleogeographic polygons as a percentage of all marine fossils at the time interval, and in contrast, the marine fossils inconsistency ratio, by the marine fossils not within shallow marine paleogeography as a percentage of all marine fossils. Similarly, the terrestrial fossils consistency ratio is defined by the terrestrial fossils within landmass, mountain or ice sheet feature as a percentage of all terrestrial fossils at the time interval and

the terrestrial fossils inconsistency ratio, by terrestrial fossils within shallow marine paleogeographic polygons as percentage of all terrestrial fossils at the time interval. Heine et al. (2015) applied a similar metric to evaluate global paleoshoreline models since the Cretaceous.

This test shows relatively high consistency between fossil paleo-environments and the underlying

paleogeographic features (Fig. 6). The results since the Cretaceous are similar to that of Heine et al. (2015). In this study, the consistency ratios of marine and terrestrial fossils during 402-2 Ma both are generally over 50%, with an average of 74.8% (marine fossils, Fig. 6a, shaded area) and 77.1% (terrestrial fossils, Fig. 6b, shaded area) but both accompanying strong fluctuations over time. Only at the time interval of 402-380 Ma, the terrestrial fossils consistency ratio drops to approximately 20.0%,

but this result is not reliable because there are only 18 terrestrial fossil collections available for this time interval.

The inconsistent marine and terrestrial fossils are used to improve marine-terrestrial boundaries in the paleogeographic maps according to the rules outlined in the Method section. Subsequently, the modified paleogeographies are tested using the same fossils. The results show the consistency ratios of marine and terrestrial fossils increased to average 97.1% (marine fossils, Fig. 6a, black line) and



average 85.9% (terrestrial fossils, Fig. 6b, black line) respectively after paleogeographies are modified
and the overall fossils, rising from average 76.9% before modification (Fig. 6c, shaded areas) to
average 96.1% after modification (Fig. 6c, black lines). Marine fossils (Fig. 6a, black lines) show better
final consistency than terrestrial fossils (Fig. 6b, black lines), mainly because marine fossils records are
less sparse than terrestrial fossils through time (Fig. 6d).

The sums of terrestrial and marine fossil collections change significantly over time (Fig. 6d), for
example, more than 4200 in total within 269-248 Ma but less than 50 in 37-29 Ma. These variations
could be due to the spatiotemporal sampling bias and incompleteness of the fossil record (Benton et al.,
2000; Benson and Upchurch, 2013; Smith et al., 2012; Valentine et al., 2006), biota extinction and
recovery (Hallam and Wignall, 1997; Hart, 1996) or our temporal selection criterion. In addition, the
differences in the duration of geological time subdivisions lead to some time-intervals having shorter
time spans that contain fewer fossil records. Specifically, marine fossils are generally more common
than terrestrial fossils (Fig. 6d) as shallow marine environments can provide conditions that are more
favorable to the preservation of biological organisms. As for the time intervals during which fossil data
is scarce, paleobiology data is of limited use in improving paleogeography. For instance, there are less
than 300 fossil collections in total in the time interval of 380-359 Ma mainly due to the late Devonian
mass extinction (McGhee, 1996). However, additional records in the future will increase the usefulness
of the Paleobiology Database in such instances.

**4.2 Improved Global Reconstructed Paleogeography**

Based on the testing results of the time intervals, we can improve the marine-terrestrial boundaries in
the global reconstructed paleogeographic maps using the approach described in Method section. The
resulting improved global paleogeographic maps since the Devonian are presented in Figure 7.
Although the modifications make the areal change minimally with regards to a global context, the
resulting paleogeographies can provide us more accurate marine-terrestrial boundaries that would be
important to generate precise paleoshorelines and therefore help constrain past changes in sea level and
long-wavelength topography.

[


We subsequently calculate the area covered by each paleogeographic feature as a percentage of the
Earth's total surface area at each time interval (Fig. 8b), using the HEALPix pixelization method that
results in equal sampling of data on a sphere (Górski et al., 2005) and therefore equal sampling of
surface areas. This method effectively excludes the effect of overlaps between paleogeographic
geometries. Using the resulting percentages of the paleogeographic features at each time interval, we
determine their surface areas on Earth (Fig. 8a) and their percentages accounting for the Earth's total
surface area (Fig. 8b) for each time interval between 402 and 2 Ma.




As a result, the areas of landmass, mountain and ice sheet generally indicate increasing trends, while shallow marine and deep ocean areas show decreasing trends through time (Fig. 8). Overall, the computed areas are sequentially becoming larger in the order of ice sheet (average 1.0% of Earth surface), mountain (3.4%), shallow marine (14.2%), landmass (21.3%) and deep ocean (60.1%). Only

in the time interval of 323-296 Ma, landmass and shallow marine areas are nearly equal at about 14.0%, and only during 359-285 Ma, ice sheet areas exceed mountain areas but ice sheets only exist during 380-285, 81-58, and 37-2 Ma. With Pangea formation in the latest Carboniferous or the Early Permian and breakup initiation in the Early Jurassic (Blakey, 2003; Domeier et al., 2012; Lenardic, 2016; Stampfli et al., 2013; Vai, 2003; Veever, 2004; Yeh and Shellnutt, 2016), these paleogeographic

features areas change remarkably over time (Fig. 8). During 323-296 Ma (Late Carboniferous-the earliest Permian), landmasses reached their smallest area and subsequently underwent a rapid increase until they peaked at 26.7% in 224-203 Ma (Late Triassic). In contrast, ice sheets reached their largest area at that time. In the Early Jurassic of Pangea breakup, landmass areas rapidly decreased from 26.7% in 224-203 Ma to 24.6% in 203-179 Ma but shallow marine areas significantly increased by 3.7%.


**4.3 Global Continental Flooding History**

We calculate the global flooding ratio of continental crust from 402 to 2 Ma (Fig. 9a, blue) by dividing the shallow marine area (Fig. 8a, lightblue) by the total continental area (inclusive of shallow marine, landmass, mountain and ice sheet; Fig. 9b, blue). The continental flooding ratios rapidly decrease from about 45.2% in the Late Devonian to 27.7% in 224-203 Ma of the Late Triassic, after that it peaks, with frequent fluctuations, at 41.8% in 94-81 Ma of the Late Cretaceous. That is then followed by a quick

decrease again until it reaches the lowest point at 27.6% in 11-2 Ma.

**5 Discussions**
**5.1 Flooding history, global sea level changes, and assembly and breakup of Pangea**

The continental flooding history we calculate between 402 and 2 Ma shows trends that are generally similar to global long-term sea level change (Haq et al., 1987; Haq et al., 2002; Müller et al., 2008; Fig. 9a). The eustatic sea level of Haq et al. (1987) and Haq et al. (2002) are inferred from the flooding ratios. Continental flooding decreases during Pangea amalgamation from the late Devonian until the Late Carboniferous, which is also reflected by low eustatic sea levels. Starting from the Early Jurassic

with the breakup of Pangea, continental flooding is increased rapidly until the Late Cretaceous when it peaked at about 42.0%. This rapid increase could be explained by a reduction of ocean volume basin associated with a decrease of the average age of the ocean floor and an increase in mid-ocean ridge length during Pangea breakup (Hays and Pitman, 1973; Müller et al., 2008; Müller et al., 2016; Van



Avendonk et al., 2016). Since the Late Cretaceous, global continental flooding rapidly decreases again simultaneously with global sea level falling, which primarily reflects the increasing age of the ocean floor (Miller et al., 2005; Müller et al., 2008). Overall, the changes of the global continental flooding during 402-2 Ma are consistent with global long-term sea level changes.

**5.2 Emerged land areas, total continental areas, continental growth models, $^{87}Sr/^{86}Sr$ of ocean water, and assembly and breakup of Pangea**

We calculate the global emerged land areas since the Devonian from the improved global reconstructed paleogeographic features of landmass, mountain and ice sheet as percentages of the Earth's surface area (Fig. 9b, red). The results generally indicate ongoing increasing continental emergence varying from about 21.0% in the Devonian to nearly 30.0% in the Neogene. Emerged land areas were slightly larger between 58 and 2 Ma (up to 30%) and between 224 and 203 Ma (27.7%) than at present (27.5%). In contrast, the evolution of the emerged land areas is inverse to the global long-term sea level changes during this time (Fig. 9a), as expected.

Similarly, the total continental areas from 402 to 2 Ma are calculated from the improved global reconstructed paleogeographies including shallow marine, landmass, mountain and ice sheet. They show a sustained increase of continental areas, rising from 37.7% in 402-380 Ma to 41.1% in 11-2 Ma (Fig. 9b, green). Before the breakup of Pangea is initiated in the Late Triassic, the total continental areas generally remain constant at an average of about 38.0% of Earth's total surface area. Continental areas then increase between 203 and 179 Ma and peak at about 44.0% in the Early Neogene, followed by a sharp decrease ending up 41.1% in 11-2 Ma. The total continental areas from the latest Early Paleogene to the earliest Neogene were larger as compared to present-day continental area 42.5% of Earth's total surface area (Schubert and Reymer, 1985). Additionally, the differences between the total continental areas and emerged land areas over time indicate large submerged continental areas since the Late Paleozoic, which comprised an average of 14.0% of Earth's surface area.

A variety of continental growth models have been proposed (e.g. Armstrong, 1981; Veizer and Jansen, 1979). Flament et al. (2013) present an integrated model to investigate the emerged area of continental crust as a function of continental growth. They predicted that the emerged land areas constantly increased from between ~21% and ~24% (CGM) at 402 Ma to 27% at 2 Ma, and the total continental area from between ~33% and ~38%(CGM) at 402 Ma to 42% at 2 Ma. Their results are generally consistent with the percentages of emerged land areas and total continental areas calculated in this study using paleogeographic features, despite some high frequency fluctuations in Early Jurassic and Late Cretaceous (Fig. 9b) indicated from our results.

The increase in the strontium isotope ratio ($^{87}Sr/^{86}Sr$) recorded in marine carbonates was previously thought to reflect continental growth (e.g. Taylor and McLennan, 1985; Veizer and Jansen, 1979). The input of high radiogenic strontium from the continents to the oceans depends on the area of emerged



land and continental relief (Godderis and Veizer, 2000). Our calculated emerged land areas from

Triassic to present show a similar changing trend with the evolution of $^{87}Sr/^{86}Sr$ of ocean water
(McArthur et al., 2012) although not for the older times (Fig. 9b). In contrast, the continental area in
the entire timeframe appears not to indicate obvious consistency with the evolution of $^{87}Sr/^{86}Sr$ of
ocean water. Therefore, we confirm that $^{87}Sr/^{86}Sr$ in ocean water may have good correlation with
emerged land area (Godderis et al., 2014; van der Meer et al., 2017) rather than continental crust area

(Flament et al., 2013).

### 6 Conclusions

Our study highlights the flexibility of digital paleogeographic models linked to state-of-the-art plate
tectonic reconstructions in order to better understand the interplay of continental growth and eustasy,

with wider implications for understanding Earth's paleotopography, ocean circulation, and the role of
mantle convection in shaping long-wavelength topography. We present a workflow that enables the
construction of paleogeographic maps with flexible spatial and temporal resolutions, while also
becoming more testable and expandable with the incorporation of new paleo-environmental datasets.
We also develop an approach to improve paleogeographic maps, especially the terrestrial-marine

boundaries, using paleobiology data.

Comparing the continental flooding history since the late Devonian inferred from our improved global
reconstructed paleogeographies with global long-term sea level change indicates that global continental
flooding ratios are consistent with global sea level change. We calculate the global emerged land areas

during 402-2 Ma from the improved global reconstructed paleogeographies. The evolution of the
emerged land areas is inverse to global sea level changes during the time, as expected.

The total continental areas during 402-2 Ma, calculated from our improved reconstructed
paleogeographies, shows good consistency with predictions of the long-term evolution of emerged land.

The emerged land area from Triassic to present shows similar evolution with $^{87}Sr/^{86}Sr$ record of ocean
water, while the total continental crust area does not. This confirms that the change of $^{87}Sr/^{86}Sr$ in
ocean water through time reflects fluctuations in emerged land area rather than in continental crust area.

### Supplementary data

We provide the shapefiles of the global paleogeographic maps during 402-2 Ma improved using

paleobiology, the GeoTiff files of all these maps, the paleobiology data in shapefile used in this study,
an animation for the improved global paleogeographic maps, and a README file outlined the
workflow of this study. All supplementary material can be downloaded from the link
(https://www.dropbox.com/sh/jzsrnnpgxrdzpaa/AAAShE5xhDxr1hmKpoBaa1G4a?dl=0).

### Acknowledgements



This work was supported by Australian Research Council grants ARC grants IH130200012 (RDM, SZ),
      DE160101020 (NF) and SIEF RP 04-174 (SW). We thank Julia Sheehan and Logan Yeo for digitizing
      these paleogeographic maps, and John Cannon and Michael Chin for help with GPlates and pyGPlates.

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




**Table 1. Time scale of Golonka et al. (2006)'s paleo-maps since the Early Devonian, their numerical equivalents as defined by Sloss (1998), and corresponding reconstruction times.**

| Era | Epoch | Nominal Age | Sloss (1988) Start Age (Ma) | Sloss (1988) End Age (Ma) | Reconstruction Time (Ma) |
|---|---|---|---|---|---|
| Cenozoic | Tortonian-Gelasian | Late Tejas III | 11 | 2 | 6 |
| | Burdigigalian-Serravallian | Late Tejas II | 20 | 11 | 14 |
| | Chattian-Aquitanian | Late Tejas I | 29 | 20 | 22 |
| | Priabonian Rupelian | Early Tejas III | 37 | 29 | 33 |
| | Lutetian-Bartonian | Early Tejas II | 49 | 37 | 45 |
| | Thanetian-Ypresian | Early Tejas I | 58 | 49 | 53 |
| Mesozoic | Late Cretaceous-earliest Paleogene | Late Zuni IV | 81 | 58 | 76 |
| | Late Cretaceous | Late Zuni III | 94 | 81 | 90 |
| | Early Cretaceous-earliest Late Cretaceous | Late Zuni II | 117 | 94 | 105 |
| | Early Cretaceous | Late Zuni I | 135 | 117 | 126 |
| | latest Late Jurassic-earliest Early Cretaceous | Early Zuni III | 146 | 135 | 140 |
| | Middle Jurassic-Late Jurassic | Early Zuni II | 166 | 146 | 152 |
| | Middle Jurassic | Early Zuni I | 179 | 166 | 169 |
| | Early Jurassic-earliest Middle Jurassic | Late Absaroka III | 203 | 179 | 195 |
| | Late Triassic-earliest Jurassic | Late Absaroka II | 224 | 203 | 218 |
| | Early-earliest Late Triassic | Late Absaroka I | 248 | 224 | 232 |
| Paleozoic | Late Permian | Early Absaroka IV | 269 | 248 | 255 |
| | Early Permian | Early Absaroka III | 285 | 269 | 277 |
| | latest Carboniferous-earliest Permian | Early Absaroka II | 296 | 285 | 287 |
| | Late Carboniferous | Early Absaroka I | 323 | 296 | 302 |
| | Early Carboniferous | Kaskaskia IV | 338 | 323 | 328 |
| | lastest Devonian-Early Carboniferous | Kaskaskia III | 359 | 338 | 348 |
| | Middle-Late Devonian | Kaskaskia II | 380 | 359 | 368 |
| | Early-Middle Devonian | Kaskaskia I | 402 | 380 | 396 |

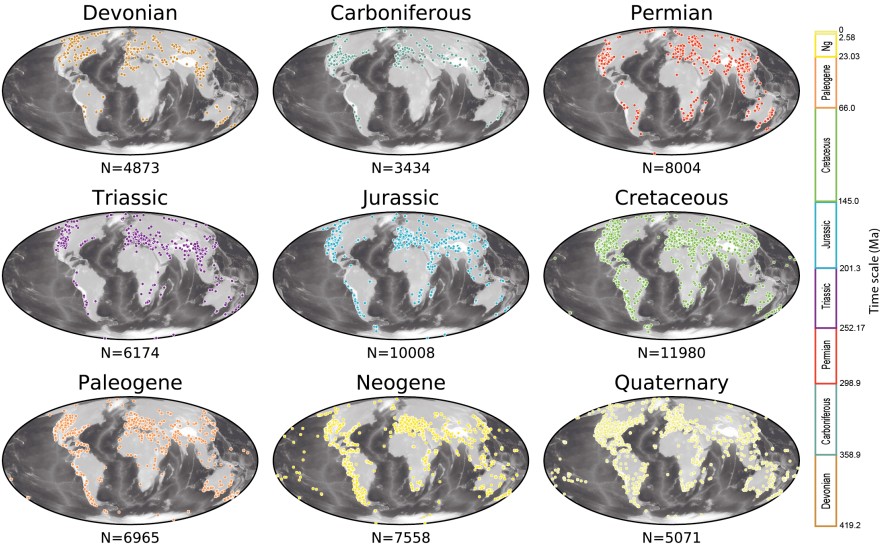


**Fig. 1. Global distributions and numbers of fossil collections since the Devonian. The greyscale background shows global present-day topography ETOPO1 (Amante and Eakins, 2009) with lighter shades corresponding to increasing elevation. Fossil collections from the Paleobiology Database are colored according following the standard used by the International Commission on Stratigraphy.**





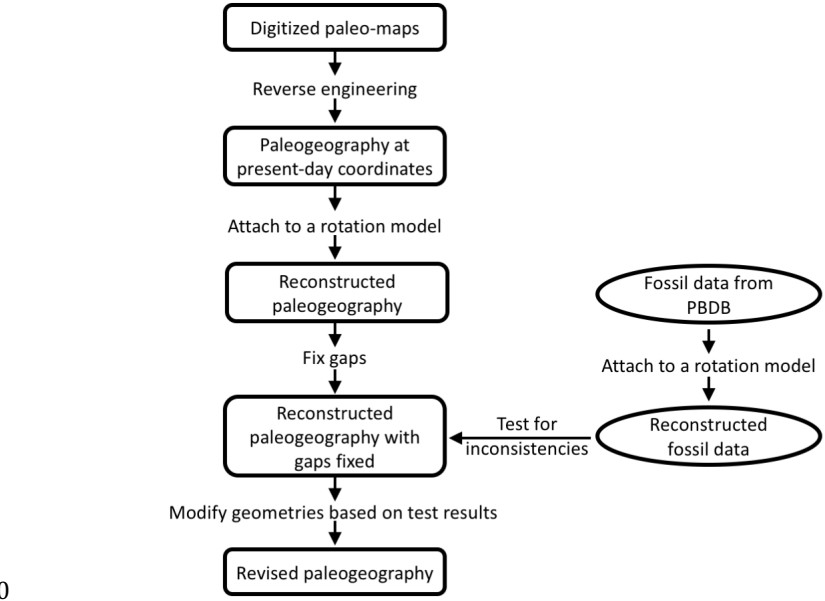


**Fig. 2. Workflow used to reverse-engineer paleogeographic reconstructions and revise them using paleobiology data. PBDB: Paleobiology Database.**


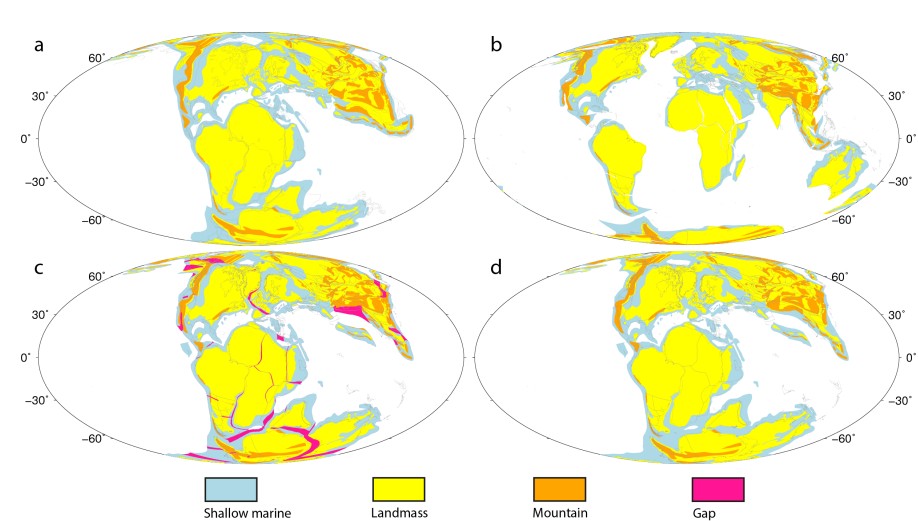

**Fig. 3. (a) Reconstructed paleogeography from Golonka et al. (2006) at 126 Ma. (b) Global paleogeography at 126 Ma in present-day coordinates. (c) Global paleogeography at 126 Ma reconstructed using the plate motion model of Matthews et al. (2016). Gaps are highlighted in pink. (d) Global paleogeography at 126 Ma reconstructed using the reconstruction of Matthews et al. (2016) with gaps fixed by filling with adjacent paleo-environment attributes. Gray lines indicate reconstructed present-day coastlines and terrane boundaries. Mollweide projection with 0°E central meridian.**






**Table 2. A lookup table for classifying fossils indicating different paleo-environments into marine or terrestrial settings and their corresponding paleogeographic types presented in Golonka et al. (2006). Terrestrial fossil paleo-environments correspond to paleogeographic features of landmasses, mountains or ice sheets, and marine fossil paleo-environments to shallow marine environments.**

| Marine | | Terrestrial | |
|---|---|---|---|
| **Paleogeography** | **Fossil Paleoenvironments** | **Paleogeography** | **Fossil Paleoenvironments** |
| | marine indet. | | terrestrial indet. |
| | carbonate indet. | | fluvial indet. |
| | peritidal | | alluvial fan |
| | shallow subtidal indet. | | channel lag |
| | open shallow subtidal | | coarse channel fill |
| | lagoonal/restricted shallow subtidal | | fine channel fill |
| | sand shoal | | channel |
| | reef, buildup or bioherm | | wet floodplain |
| | perireef or subreef | | dry floodplain |
| Shallow marine environments | intrashelf/intraplatform reef | Landmasses/Mountains | floodplain |
| | platform/shelf-margin reef | | crevasse splay |
| | slope/ramp reef | | levee |
| | basin reef | | mire/swamp |
| | deep subtidal ramp | | fluvial-lacustrine indet. |
| | deep subtidal shelf | | delta plain |
| | deep subtidal indet. | | fluvial-deltaic indet. |
| | offshore ramp | | lacustrine - large |
| | offshore shelf | | lacustrine - small |
| | offshore indet. | | |
| | slope | | |

| | basinal (carbonate) | | pond |
|---|---|---|---|
| | basinal (siliceous) | | crater lake |
| | marginal marine indet. | | lacustrine delta plain |
| | coastal indet. | | lacustrine interdistributary bay |
| | estuary/bay | | lacustrine delta front |
| | lagoonal | | lacustrine prodelta |
| | paralic indet. | | lacustrine deltaic indet. |
| | delta plain | | lacustrine indet. |
| | interdistributary bay | | dune |
| | delta front | | interdune |
| | prodelta | | loess |
| | deltaic indet. | | eolian indet. |
| | foreshore | | cave |
| | shoreface | | fissure fill |
| | transition zone/lower shoreface | | sinkhole |
| | offshore | | karst indet. |
| | submarine fan | | tar |
| | basinal (siliciclastic) | | spring |
| | deep-water indet. | | |

| | | Ice sheets | glacial |
|---|---|---|---|




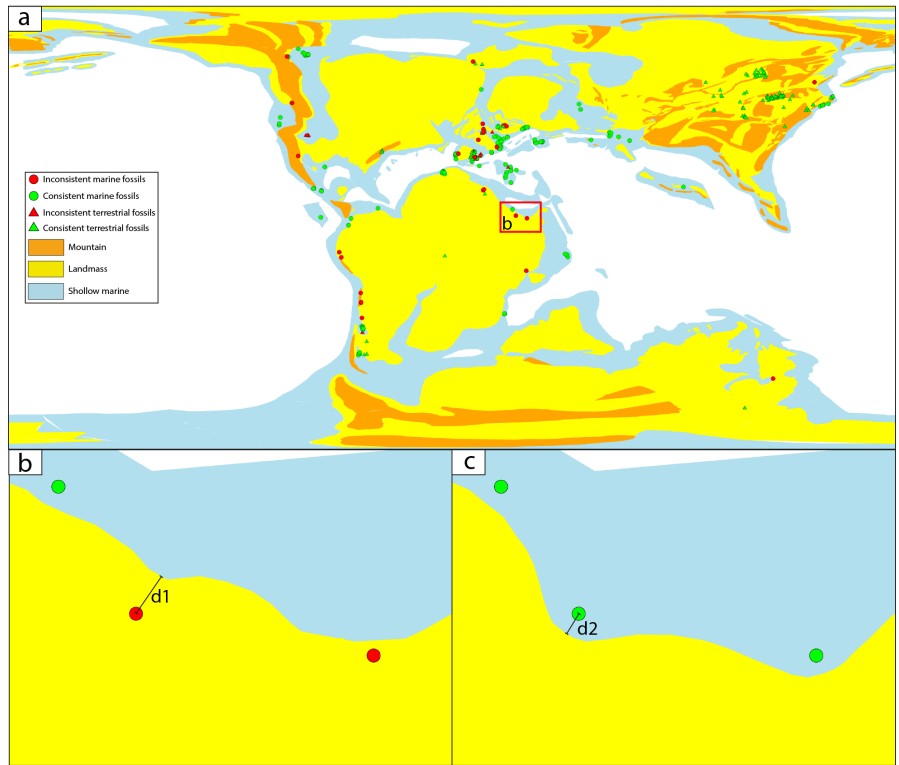


**Fig. 4. (a) Global paleogeography at 126 Ma (Golonka et al., 2006) reconstructed using the model of Matthews et al. (2016) tested against terrestrial and marine fossil collections recorded in the Paleobiology Database. (b) The zoomed-in area of the small box in (a) highlights inconsistent marine fossils (red points). d1 ≤ 100 km is the distance between the inconsistent marine fossil and its nearest terrestrial-marine boundary. (c) illustrates how a terrestrial-marine boundary is modified based on inconsistent fossils, the terrestrial-marine boundary is shifted until the two marine fossils are consistent with the underlying paleogeography and at the same time keep about 20 km distance from their nearest boundary, d2 ≈ 20 km is the distance between the fossil used and its nearest terrestrial-marine boundary shifted.**





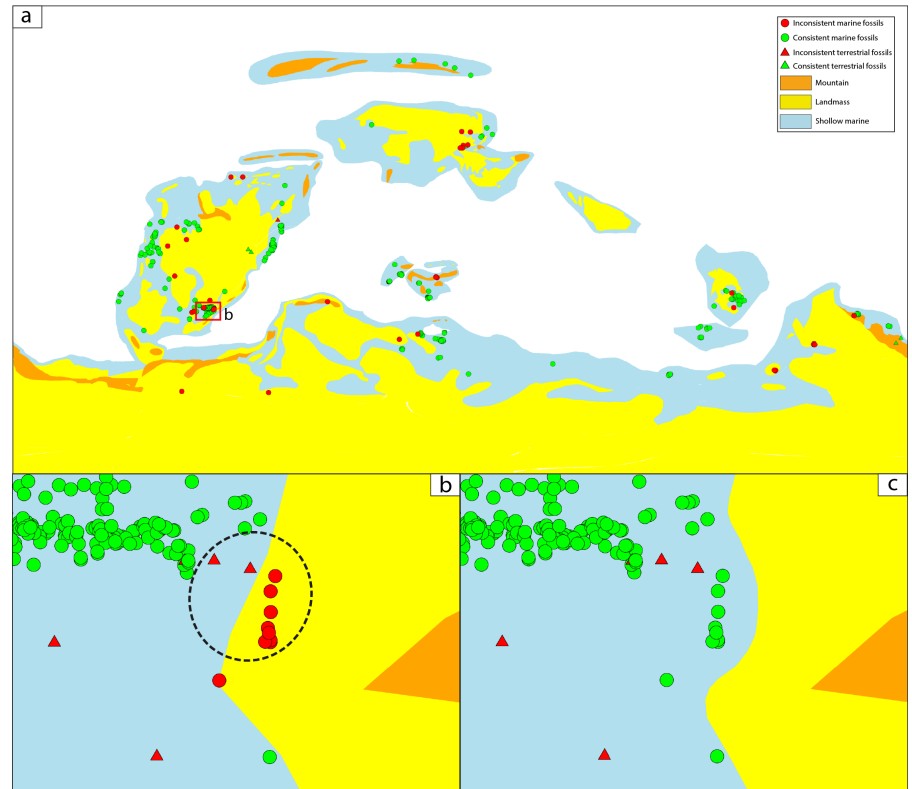

**Fig. 5. (a) Global paleogeography reconstructed at 396 Ma tested by terrestrial and marine fossils. (b) Zoomed-in area of the small box in (a). The fossils in the black circle are considered as a cluster, in which over 50% of fossils indicate shallow marine environment, therefore, the whole cluster is interpreted as shallow marine. (c) Illustrates how the terrestrial-marine boundary is shifted to be reconciled with fossil collections.**






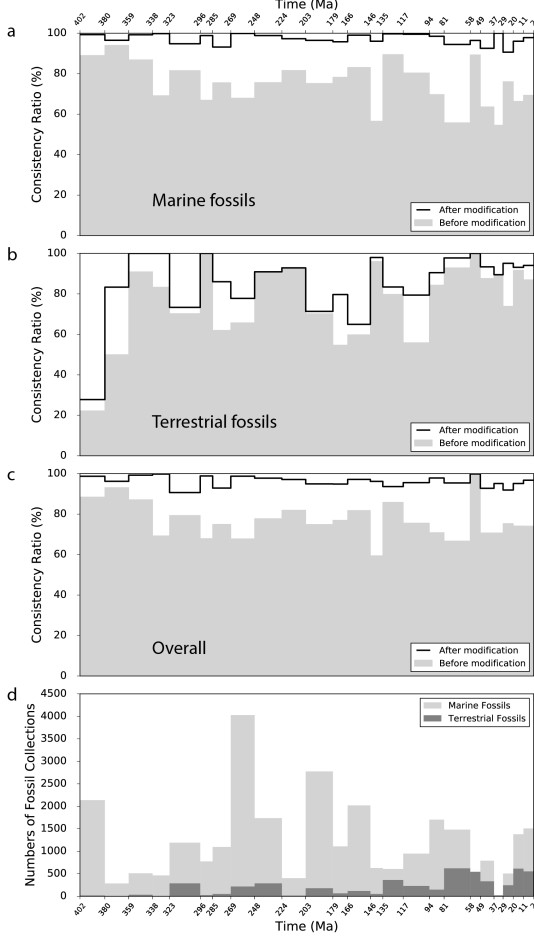

**Fig. 6. (a) Global consistency ratios between marine fossils and underlying paleogeographies before (shaded areas) and after (black lines) modification based on fossils for each time interval between 402 and 2 Ma. (b) Global consistency ratios between terrestrial fossils and underlying paleogeographies before (shaded areas) and after (black lines) modification for each time interval between 402 and 2 Ma. (c) Global consistency ratios between total fossils (marine and terrestrial) and underlying paleogeographies before (shaded areas) and after (black lines) modification for each time interval between 402 and 2 Ma. (d) Numbers of terrestrial (shaded in dark grey) and marine (shaded in light grey) fossil collections for each time interval between 402 and 2 Ma.**







Fig. 7. Global paleogeographies from 402 to 2 Ma reconstructed with the plate reconstruction of Matthews et al. (2016) and improved using paleobiology data. Black toothed lines indicate subduction zones, and other black lines denote mid-ocean ridges and transforms. Gray outlines delineate reconstructed present-day coastlines and terranes. Mollweide projection with 0°E central meridian.






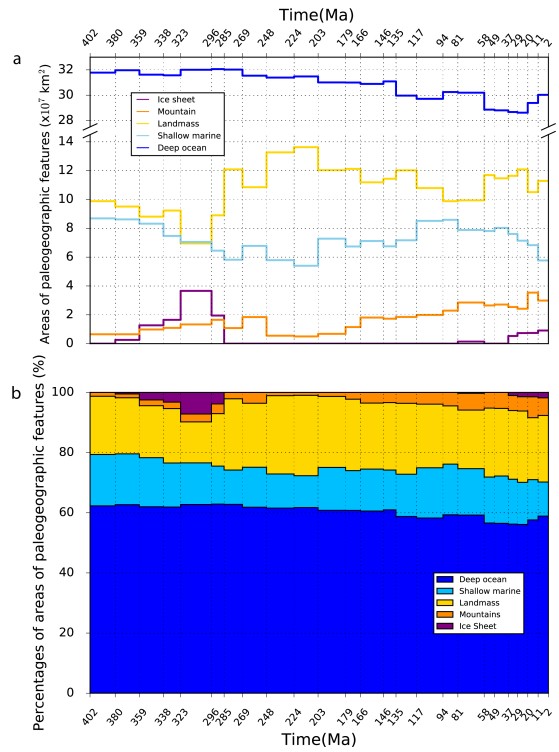

**Fig. 8. (a) Global paleogeographic feature surface areas from 402 to 2 Ma. (b) Global paleogeographic**

**feature areas as percentages of the Earth's total surface area at each time interval from 402 to 2 Ma.**



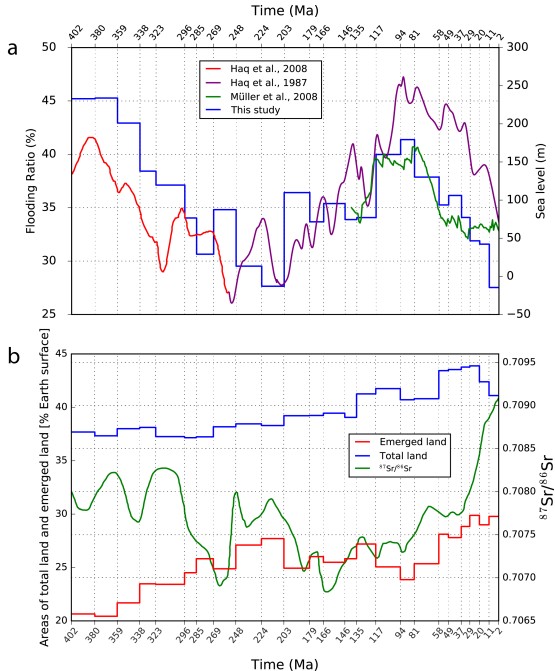

**Fig. 9. (a) Global continental flooding ratio since the Devonian (blue) and global sea level from Haq et al.**


**(1987) (purple), Haq et al. (2008) (red) and Müller et al. (2008) (green). (b) Total continental areas (blue) and emerged land areas (red) as a percentage of Earth's surface area. $^{87}$Sr/$^{86}$Sr record of ocean water of McArthur et al. (2012) (Green). Total continental area comprises shallow marine, landmass, mountain and ice sheet. Emerged land comprises landmass, mountain and ice sheet. Flooding ratio is defined as shallow marine area divided by the total continental area.**
