# Peer review of "Improving global paleogeography since the late Paleozoic using paleobiology"

_Biogeosciences, 2017_

## Referee Comment (RC1) · Anonymous Referee #1 · 3 May 2017

The comment was uploaded in the form of a supplement:
http://www.biogeosciences-discuss.net/bg-2017-94/bg-2017-94-RC1-supplement.zip

---

## Referee Comment (RC2) · Anonymous Referee #2 · 3 May 2017

The authors attempt to produce a flexible, digital representation of Earth's plates through most of the Phanerozoic. This representation should allow testing paleogeographic features of the original dataset against other datasets, adopting different rotation models as used in the original dataset, among other things. The authors then use a comparison of their original distributions of land and sea to that implied by the distribution of fossil organisms, to get a more accurate picture of the distributions of land and sea through Earth's history. These 'improved' distributions are then used for various comparisons with eustatic sealevel curves and measures for continental weathering.

Although the attempt to build a flexible model of Earth's plate movements through time is fine and useful, most of the subsequent comparisons are, in my view, redundant, insufficiently interpreted and discussed. Also the methods section needs improvements. In the present state I can only recommend to reject the manuscript, and to encourage

the authors to focus on the core of their work (the model), to improve the methods section, and revamp their 'testing' and their discussion.

Detailed comments by line number:

106-108, there is another important bias in the PBDB: the uneven entry of fossil data.

116-117, repetition

145-147, I have the feeling that the authors are trying to explain here which environmental types have gone into the gaps and overlaps, but I failed to understand it.

155-159, here the authors sometimes talk about 'fossil collections' and sometimes about 'fossils', though my impression is that they always mean 'fossil collections' – please be consistent here and throughout the ms in general.

187-190, unclear how it was decided which 'fossils' (by which the authors presumably mean 'fossil collection site') are included in such a cluster and which aren't. It is important to make clear how the boundaries of these clusters are drawn.

235-243, this entire test is redundant: if you're adjusting the land-sea boundary in such a way that most inconsistencies are removed, of course does your 'consistency index' improve.

Paragraph 245-257, it is not clear to me what the authors are getting at with this paragraph. They discuss various biases and inhomogeneities of the fossil data, but neither do they apply a coherent test to the problem, nor do they reach any conclusion (except perhaps for "fewer fossils = fewer possibilities for adjustments", but this again is trivial).

245-249, as for lines 106-108, uneven entry of data is another potential bias.

249-251, "shorter time spans contain fewer fossils" – it might be interesting to systematically test the fossil dataset for this.

253, "biological organisms" – organisms are biological by definition

264-267, here I was wondering how much of the "areal change" might relate to the gap filling and overlap removal that the authors have done to fit the plate reconstructions. In their lines 144-145 they wrote that the total areal variations ranged from 5.8 to -2.7%. A comparison of these values through time to the extent of area change through time (or something along these lines) might provide valuable insights here.

281ff, unless I've overlooked it, there is a step missing here in the explanation of the method. So far, the authors explained that in their adjustments, they exchanged 'land' for 'sea' and vice versa. But now they start discussing the quantification of different habitat types (shallow vs. deep sea, mountains vs. low lands etc.). Does this mean that when the land-sea boundary was shifted, for example, the 'new sea area' was assigned the habitat type of the fossil collection that caused the change? For example, has an area previously classified as 'mountain' sometimes been replaced by 'shallow marine' and sometimes by 'deep marine'? If so, this needs to be explained in the Methods section.

310ff, this whole paragraph seems redundant. It is pretty obvious to any earth scientist that continental flooding and eustatic sealevel changes are linked. Not only is it obvious that eustatic sealevel changes cause continental flooding (what else should it be?); to make matters worse, the eustatic sealevel curves are inferred from the continental flooding history as recorded in the sedimentary record so you might be looking at circularity here.

332, the difference between 27.7% and 27.5% isn't really great, isn't it? The authors should be a little more cautious about the errors in their own model. Could this difference of 0.2% again result from their gap filling procedure? Or could it be related to the inconsistencies in their 'improved paleogeographies'? In their lines 238-241 they write that even their 'improved paleogeographies' are still 3-5% inconsistent, which is a lot more than the 0.2% difference mentioned above. I recommend that the authors assess these inherent errors in their model (gap filling and 'consistency' index) and then discuss only variations that exceed those errors.

341, 3% of the world's continental area has disappeared in the Neogene? Where did it go?

350-351, the abbreviation CGM is not explained (and perhaps not necessary?)

363, I find it dubious to 'confirm that Sr isotope ratios have a good correlation with emerged land areas' when there is no such correlation in the Paleozoic. Doesn't this rather indicate that there may be something fundamentally wrong with this correlation? I have no solution to the problem, but it seems more scientifically to me to point out such inconsistencies rather than to uncritically reiterate some lukewarm 'conventional wisdom'.

366ff, the 'Conclusions' nicely sum up the good parts and the problems of this study. The first paragraph outlines the good part, the flexible, digital plate model that could surely be of use for a wide range of earth scientists. The second paragraph discusses the redundant correlation between emerged land and eustatic sealevel changes, and the third paragraph again 'confirms' a correlation between Sr isotopes and emerged land, which apparently doesn't exist in the Paleozoic.

Table 1. why is this awkward Sloss 1988 timetable used? As far as I can tell, it applies to the US only, and connecting it to the accepted ICS and GSA timescales and to the periods, series and stages that have been used by geologists for more than 100 years is confusing. Avoid this, it is of no use for geologists and paleontologists.

Table 2, I had difficulties relating this table to what's written in the manuscript. The table distinguishes three paleogeographies (shallow marine, landmass/mountain, ice sheet), whereas in the text and fig 8 five distinctions are made (shallow marine, deep marine, land masses, mountains, ice sheets). Please be consistent here.

Figure 5. colors and shapes are not explained; perhaps refer to fig. 4? And I presume you mean "fossil collection sites" rather than "fossils"? I don't see any fossils in this figure.

[Figure]

---

## Author Comment (AC1) · 3 Jul 2017

Please find attached the compressed file.

Please also note the supplement to this comment:
https://www.biogeosciences-discuss.net/bg-2017-94/bg-2017-94-AC1-supplement.zip

---

## Author Comment (AC2) · 3 Jul 2017

**Reviewer:** The authors attempt to produce a flexible, digital representation of Earth's plates through most of the Phanerozoic. This representation should allow testing paleogeographic features of the original dataset against other datasets, adopting different rotation models as used in the original dataset, among other things. The authors then use a comparison of their original distributions of land and sea to that implied by the distribution of fossil organisms, to get a more accurate picture of the distributions of land and sea through Earth's history. These 'improved' distributions are then used for various comparisons with eustatic sea level curves and measures for continental weathering. Although the attempt to build a flexible model of Earth's plate movements through time is fine and useful, most of the subsequent comparisons are, in my view, redundant, insufficiently interpreted and discussed. Also the methods section needs improvements. In the present state I can only recommend to reject the manuscript, and to encourage the authors to focus on the core of their work (the model), to improve the methods section, and revamp their 'testing' and their discussion.

**Authors:** We thank the reviewer for his/her constructive review that will guide our revision of the manuscript. We will amend the paleogeographic model, give more detail in the Method section, and change the tests carried out on the paleogeographies using paleobiology. We will delete the comparison between continental flooding curves and published sea level fluctuations as there may be some circularity in this comparison, and the comparison between emerged land area, total land area and the evolution of strontium isotopes of marine carbonates. Instead, we will compare our flooded continental area curve to previously published ones (see Fig. 1 below). We will estimate the terrestrial and oceanic areal change due to filling gaps and modifying the coastline locations and the paleogeographic geometries over time (see Fig. 2 below), test the marine fossil collection dataset used in this study for fossil abundances over time with two different time scales (see Fig.3 below), and discuss the limitations of the workflow we developed in this study.

**Reviewer:** Detailed comments by line number: 106-108, there is another important bias in the PBDB: the uneven entry of fossil data.

**Authors:** We agree and will add this to the sentence in the revision.

**Reviewer:** 116-117, repetition

**Authors:** We will rewrite this sentence in the revision.

**Reviewer:** 145-147, I have the feeling that the authors are trying to explain here which environmental types have gone into the gaps and overlaps, but I failed to understand it.

**Authors:** We will delete this sentence to avoid any confusion.

**Reviewer:** 155-159, here the authors sometimes talk about 'fossil collections' and sometimes about 'fossils', though my impression is that they always mean 'fossil collections' – please be consistent here and throughout the ms in general.

**Authors:** Yes, they all mean 'fossil collections'. This will be corrected throughout the manuscript.

**Reviewer:** 187-190, unclear how it was decided which 'fossils' (by which the authors presumably mean 'fossil collection site') are included in such a cluster and which aren't. It is important to make clear how the boundaries of these clusters are drawn.
**Authors:** In our revised version of the maps, we will only use marine fossil collections to improve paleo-coastline locations and the paleogeographic geometries (see Fig. 4, 5, 6 below), because the coastlines on the paleo-maps used in this study represent maximum transgression surfaces, so this is not the case anymore.

**Reviewer:** 235-243, this entire test is redundant: if you're adjusting the land-sea boundary in such a way that most inconsistencies are removed, of course does your 'consistency index' improve.
**Authors:** We will delete the test of modified paleogeography with paleobiology, and will only present the test of unmodified paleogeography (see Fig. 6 below).

**Reviewer:** Paragraph 245-257, it is not clear to me what the authors are getting at with this paragraph. They discuss various biases and inhomogeneities of the fossil data, but neither do they apply a coherent test to the problem, nor do they reach any conclusion (except perhaps for "fewer fossils = fewer possibilities for adjustments", but this again is trivial).
**Authors:** We will apply a test on the marine fossil collection dataset used in this study for fossil abundances over time with two different time scales: ICS2016 and Golonka (2000) (see Table 1 below), and we will revise this paragraph, delete the trivial part, present the result (see Fig. 3 below) and discuss it in the Discussions section.

**Reviewer:** 245-249, as for lines 106-108, uneven entry of data is another potential bias.
**Authors:** We will add this in the revision.

**Reviewer:** 249-251, "shorter time spans contain fewer fossils" – it might be interesting to systematically test the fossil dataset for this.
**Authors:** We will test the dataset used in this study for fossil abundances over time with two different time scales: ICS2016 and Golonka (2000) (see Table 1 below), present the result (see Fig. 3 below) and discuss it in the Discussions section.

**Reviewer:** 253, "biological organisms" – organisms are biological by definition
**Authors:** We will remove "biological" in the revision.

**Reviewer:** 264-267, here I was wondering how much of the "areal change" might relate to the gap filling and overlap removal that the authors have done to fit the plate reconstructions. In their lines 144-145 they wrote that the total areal variations ranged from 5.8 to -2.7%. A comparison of these values through time to the extent of area change through time (or something along these lines) might provide valuable insights here.
**Authors:** We will estimate the areal change in two key steps of the methodology, including filling gaps and modifying the coastline locations and paleogeographic geometries, present the results (see Fig. 2 below) and explain it in the Discussions section.

**Reviewer:** 281ff, unless I've overlooked it, there is a step missing here in the explanation of the method. So far, the authors explained that in their adjustments, they exchanged 'land' for 'sea' and vice versa. But now they start discussing the quantification of different habitat

types (shallow vs. deep sea, mountains vs. low lands etc.). Does this mean that when the land-sea boundary was shifted, for example, the 'new sea area' was assigned the habitat type of the fossil collection that caused the change? For example, has an area previously classified as 'mountain' sometimes been replaced by 'shallow marine' and sometimes by 'deep marine'? If so, this needs to be explained in the Methods section.
**Authors:** We will explain this in the Method section.

**Reviewer:** 310ff, this whole paragraph seems redundant. It is pretty obvious to any earth scientist that continental flooding and eustatic sea level changes are linked. Not only is it obvious that eustatic sealevel changes cause continental flooding (what else should it be?); to make matters worse, the eustatic sealevel curves are inferred from the continental flooding history as recorded in the sedimentary record so you might be looking at circularity here.
**Authors:** We will remove this entire paragraph as indeed there could be some degree of circularity.

**Reviewer:** 332, the difference between 27.7% and 27.5% isn't really great, isn't it? The authors should be a little more cautious about the errors in their own model. Could this difference of 0.2% again result from their gap filling procedure? Or could it be related to the inconsistencies in their 'improved paleogeographies'? In their lines 238-241 they write that even their 'improved paleogeographies' are still 3-5% inconsistent, which is a lot more than the 0.2% difference mentioned above. I recommend that the authors assess these inherent errors in their model (gap filling and 'consistency' index) and then discuss only variations that exceed those errors.
**Authors:** Since we will delete the comparison between emerged land area, total land area and the evolution of strontium isotopes, this part will be removed accordingly. As suggested here, we will amend the paleogeographic model and update the test carried out on the paleogeographies using paleobiology. We will estimate the errors of two key steps in the workflow, including filling gaps and modifying the coastline locations and the paleogeography, on the terrestrial or oceanic areal change over time (see Fig. 2 below) and discuss them in the Discussions section.

**Reviewer:** 341, 3% of the world's continental area has disappeared in the Neogene? Where did it go?
**Authors:** There is an increase in mountainous areas compensating the loss in non-elevated land.

**Reviewer:** 350-351, the abbreviation CGM is not explained (and perhaps not necessary?)
**Authors:** As we will delete this entire paragraph, this will be deleted in the revision accordingly.

**Reviewer:** 363, I find it dubious to 'confirm that Sr isotope ratios have a good correlation with emerged land areas' when there is no such correlation in the Paleozoic. Doesn't this rather indicate that there may be something fundamentally wrong with this correlation? I have no solution to the problem, but it seems more scientifically to me to point out such inconsistencies rather than to uncritically reiterate some lukewarm 'conventional wisdom'.
**Authors:** We will delete the comparison between emerged land area, total land area and the evolution of strontium isotopes of marine carbonates in the revision.

**Reviewer:** 366ff, the 'Conclusions' nicely sum up the good parts and the problems of this study. The first paragraph outlines the good part, the flexible, digital plate model that could surely be of use for a wide range of earth scientists. The second paragraph discusses the redundant correlation between emerged land and eustatic sea level changes, and the third paragraph again 'confirms' a correlation between Sr isotopes and emerged land, which apparently doesn't exist in the Paleozoic.
**Authors:** Our conclusions will be amended in the revision. Thanks to the input from the reviewer.

**Reviewer:** Table 1. why is this awkward Sloss 1988 timetable used? As far as I can tell, it applies to the US only, and connecting it to the accepted ICS and GSA timescales and to the periods, series and stages that have been used by geologists for more than 100 years is confusing. Avoid this, it is of no use for geologists and paleontologists.
**Authors:** Sloss (1988) is the base of the time scale of Golonka (2000) applied to the paleogeography used in this study. We have converted the time scales of Sloss (1988) and Golonka (2000) to agree with the ICS2016 and will present them together in the table (See Table 1 below).

**Reviewer:** Table 2, I had difficulties relating this table to what's written in the manuscript. The table distinguishes three paleogeographies (shallow marine, landmass/mountain, ice sheet), whereas in the text and fig 8 five distinctions are made (shallow marine, deep marine, land masses, mountains, ice sheets). Please be consistent here.
**Authors:** We will correct this in the revision (see Table 2 below).

**Reviewer:** Figure 5. colors and shapes are not explained; perhaps refer to fig. 4? And I presume you mean "fossil collection sites" rather than "fossils"? I don't see any fossils in this figure.
**Authors:** We will replace Figure 5 by a new figure (see Fig. 5 below) in which the colours and shapes will be explained clearly. Yes, we refer to "fossil collection sites" rather than "fossils" and we will correct this throughout the manuscript.

[revised manuscript text omitted]

---

## Author Response (AR1)

General Comments:

**Reviewer:** This is an interesting paper that does an excellent job combining two disjoint data sets (plate tectonic models & paleogeography) into a cohesive synthesis. The resulting discussion of the relationship of continental flooding to sea level and to the changing ratio of strontium isotopes in the oceans through time is clearly presented. All the figures are readable and well done. The writing is patchy, but I have made numerous suggestions for the authors. This study had four principle objectives: 1) to describe the process by which the paleogeography (Golonka) developed for one plate tectonic model (Scotese) could be reverse engineered and plotted on an alternate plate tectonic model (Matthews), 2) to improve the Golonka paleogeography by adding additional constraints from the Paleobiology Database, 3) to compare the resulting estimates of continental flooding though time with published sea level curves, and finally, 4) to explain the changing ratio of strontium isotopes in the ocean with the observed patterns of continental growth and emergence. Each of these objectives was successfully met, to varying degrees. Objective 1: The new set of paleogeographic maps produced in this paper, clear demonstrates that it is possible to transfer the paleogeographic information from one set of maps (Golonka, 2006) to another set (Matthews, 2016) – as long as plate tectonic models are available for both sets of maps. However, the methodology cannot be considered to be a universal solution. As pointed out by the authors, the paleogeography and plate models are inextricably joined, and moving the paleogeography from one plate model to a another plate model inevitably results in gaps and overlaps (see Figure 3c). Unfortunately this will always be the case. It will always be necessary to laboriously "hand edit" any attempt to transfer the paleogeography from one plate model to another.

**Authors:** We thank Christopher Scotese for his constructive review and detailed suggestions that have helped us to significantly improve the manuscript. We agree with the four points he raised, to be addressed in the revision. In terms of objective 1, we agree that the methodology has some limitations and we have discussed them in the revision.

**Reviewer:** Objective 2: There are several issues here that need to be discussed. My first major point is that I am not convinced that the "revised" coastlines are a significant improvement over the original coastlines. Though, I agree that the addition of information from the Paleobiology database can, in some areas, improve the location of the coastlines, it is not clear to me that the overall result is an improvement or merely a slight modification. There are two reasons for my skepticism. Firstly, I do not know what original data was used to draw the coastlines. Therefore I do not know how much "weight" to give the Paleobiology data with regard to the original data. For example is the original coastline is based on a dozens of coastline estimates from a variety of sources, then a few additional data points from the PBDB should not be given much weight. Conversely, if the original coastline position was an educated guess based on little or no data, then the extra information from the PBDB would be very welcomed. So, simply, we don't if the changes are an improvement or not. The second reason for doubting that any improvement has

been made is to consider what the coastline drawn on the original maps actually represents. In this case, I believe the error lies with the mapmaker, not the analysis.

**Authors:** The revised paleo-coastlines are significantly different, except for a few time-interval maps where there are few paleobiology data (Please see revised Figs 4, 5, 6 and a set of maps in Supplement materials). Note that in the new tests carried out on the paleogeography with paleobiology, we only use marine fossil collections to improve paleo-coastline locations and the paleogeographic geometries because the coastlines on the paleogeographic maps used in this study represent maximum transgression surfaces. The paleogeographic atlas in the study is compiled based on gathered lithologic data, which is independent with paleobiology data. Since the original data that were used to estimate the coastlines are not available for us, it is difficult to give the weight to the paleobiology data. The coastlines drawn on the original maps represent maximum transgression surfaces and we do not know much about their errors. Instead, we have systematically estimated the errors of two key steps in the workflow, including filling gaps and modifying the coastline locations and the paleogeography (see Fig. 10 in the revision) and added their discussion in the revision (lines 341-366).

**Reviewer:** The 24 maps in this study cover ~400 million years. That means, on average, that each map represents an interval of 17 million years. It seems very unlikely that the coastline would have remained in one place for 17 million years. A more reasonable representation of the "coastline" for this long interval would have been to show it as a "zone" that was alternately marine or terrestrial. (see my Figure 1). One way to simulate this would have been to erect a 250- 500 km buffer around the coastline, and then test only the points that lied outside of the buffer. I am not suggesting that the authors do this, but rather I am suggesting that it is likely that the "discrepancies" they point out, may in fact, be perfectly OK, given the changing location of the coastline through time. In this regard, I think the manuscript would be improved if the author's pointed out this possibility and changed their wording so that it sounds less pejorative (i.e. You made mistake and now I'm going to fix it.) In fact what would be more valuable if the authors listed all the marine data points that plotted on mountain ranges or more than 500 km from the proposed coastlines, or conversely, terrestrial deposits that plotted in the deep sea (off the edges of the continents). In these cases, changes to the paleogeographic maps should certainly be made!

**Authors:** In the revised version of the maps, we only use marine fossil collections to improve coastline locations and paleogeographic geometries. We have flagged all inconsistent marine fossil collections far more than 500 km inland from the nearest coastlines with red point symbology, on each time-interval map (see a set of maps in Supplement materials).

**Reviewer:** Objective 3: Everything here looks pretty good, however there was a little graphical confusion that needs to be fixed. It is hard to argue against a positive correlation between sea level rise and continental flooding, and I am happy to see that in Figure 9A both trends track each other well. However, it is not clear which units (y-axis) apply to which curve. This should be cleared up in the Figure caption. More problematic, however, is that the fact that the figure implies that these two very different units scale together. i.e. 40% flooding = 160m rise in sea level. This is certainly not true. The cleanest solution would be to separate these two graphs, but place them one above the other.

**Authors:** We have deleted the comparison between continental flooding curves and published sea level fluctuation curves as there may be some circularity in this comparison. Instead, we only compare our flooded continental area curve to previously published ones (see revised Fig. 9).

**Reviewer:** Objective 4. The same objection raised to Figure 9a also applies to 9b. It may be necessary to separate this figure into two diagrams.
**Authors:** We have deleted the comparison between emerged land area, total land area and the strontium isotope ratio curve, so this figure has been replaced.

Additional General Comments:
**Reviewer:** The Methods Section consistently misuses verb tense.  Lines 115 – 334.  You are describing actions that you did in the past. You must use the past tense, not the present tense e.g. "They are first georeferenced" should be "They were first georeferenced. " Review all verb tenses in this section and correct.
**Authors:** Thank you. All verb tenses throughout the manuscript in the revision have been uniform using present tense.

**Reviewer:** There is a confused an improper use of the terms "fossil" and "paleobiology". No fossils were used in this paper, only fossil collections that revealed paleoenvironmental conditions, i.e., marine or terrestrial.
**Authors:** We have corrected this throughout the manuscript in the revision.

**Reviewer:** When listing ranges of dates, "Ma" should appear after each date if the dates are separated by a "and" or "to", e.g. 402 Ma and 2 Ma or 402 Ma to 2 Ma.   This is not necessary if the dates are separated by a dash, as in 402-2 Ma.
**Authors:** We have amended this in the revision.

**Reviewer:** Other specific comments regarding the text, figures or tables are given in the following section. Specific Comments by line: 016  Delete  "time-dependent global" and "Several"
**Authors:** We have deleted them in the revision.

**Reviewer:** 018  The phrase "static maps with varying temporal resolution and fixed spatial resolution" is not clear and seems redundant and should be rewritten.  Aren't all maps "static" and have a fixed "spatial resolution", i.e. "scale".  So?
**Authors:** We have rewritten this in the revision (lines 18-19).

**Reviewer:** 020 Though the authors were successful in "reverse engineering" the Golonka maps, the workflow they produced is not a general or universal solution.  Because of the idiosyncrasies of various plate tectonic reconstructions, each reverse engineered set of maps requires extensive hand editing to fix the resulting gaps and overlaps.  This will always be true.  So the claim that this new workflow fixes that problem and is a universal solution is incorrect and therefore the claim must be withdrawn or modified.
**Authors:** We agree and have modified the claim in the revision (lines 21-22). In addition, we have added the discussion of the limitations of the workflow developed in this study in the Discussions section (lines 341-366).

**Reviewer:** 022 The sentence, "Published paleogeographic . . . datasets." is not informative and should be deleted.
**Authors:** We have deleted this sentence in the revision.

**Reviewer:** 023 "fossil data" to "paleoenvironmental data".
**Authors:** We have amended this in the revision (line 23).

**Reviewer:** 023 I am not convinced that the maps were improved. See my comment above. There are some methodology problems here - both in the map making and analysis.   The best I think you can say is that "the maps were modified to be more consistent with the paleoenvironmental data from the Paleobiology database."  This statement does not imply that the resulting maps are "better". (I know this seems like nit-picking, but it actually is an important point!)
**Authors:** The paleo-maps are significantly different, except for a few time-interval maps where there have few paleobiology data (see revised Figs 4, 5, 6 and a set of maps in Supplement materials).

**Reviewer:** 039  A definition of what you mean by "paleogeography" might be appropriate here.  I favor this definition, "paleogeographic maps describe the ancient distribution of highlands, lowlands, shallow seas, and deep ocean basins".  Of the list of examples, that would disqualify Scotese (2004), but Scotese (2001 and 2004) could be substituted (see list references cited at end of review).
**Authors:** We have added the definition of "paleogeography" (lines 41-42) and corrected the references (lines 45, 500-502) in the revision.

**Reviewer:** 043  Here we go with that static .. fixed spatial resolution " business again.Why don't you just say that it is difficult to convert the maps into a digital format because of the varying map projection, different time intervals represented by the maps, and the different plate models that underlie the paleogeographic reconstructions.  I agree that there is great power to having the paleogeographic data in a digital format so you can  . . .. (examples). Yes, this is a worthwhile goal.
**Authors:** We have rewritten this part in the revision as suggested (lines 45-47). Thank you.

**Reviewer:** 052  use "these issues"
**Authors:** We have amended this in the revision (line 55).

**Reviewer:** 054  not "any plate model"  but a  "different plate model".  Your workflow is not a universal solution.   It is likely that any change in the plate model will create new gaps and overlap that will have to be fixed by hand.
**Authors:** We have changed "any plate model" to "different plate model" in the revision (lines 56-57).

**Reviewer:** 055 Try rewriting this sentence without the jargon.  "The first step was . . . "
**Authors:** We have rewritten this sentence in the revision (lines 57-59).

**Reviewer:** 058 You didn't "reverse-engineer the global maps" (whatever that means). You "restored the ancient paleogeographic boundaries back to their modern coordinates by applying the inverse of the rotation that was used to make the ancient reconstruction." More words, but more clear.
**Authors:** We have amended this claim in the revision as suggested (lines 62-64).

**Reviewer:** 060 -062 How about saying this, "Subsequently, we used information about marine and terrestrial paleoenvironments available from the Paleobiology Database to modify the location of the paleo-coastlines."
**Authors:** We have rewritten this in the revision as suggested (lines 65-67).

**Reviewer:** 068 "modelled" should be "modeled"
**Authors:** Since we have deleted the comparison between emerged land area, total land area and the evolution of strontium isotopes of marine carbonates, the whole sentence here has been deleted in the revision.

**Reviewer:** 073 "paleoenvironmental data" not "paleontological data"
**Authors:** We have modified this in the revision (line 81).

**Reviewer:** 077 see my comments about Table 1.
**Authors:** We have listed three time scales of Sloss (1988), Golonka (2000) and ICS2016 in the table (see revised Table 1).

**Reviewer:** 084 change "a plate tectonic model" to "a mysterious plate tectonic model " - just kidding! 089 not "reverse-engineer", but " restore these paleogeographies to their present-day coordinates".
**Authors:** We have amended "reverse-engineer" to "restore" in the revision (line 97).

**Reviewer:** 091 in Figure 2 of this review I show that the plate model is identical to Scotese (1997) that was published in Scotese (2004). So the sentence should read, "are based on Scotese (1997, 2004)". My plate models have been widely available – mostly through the paleomapping programs I have written (with students) – Terra Mobilis, PaleoMap-PC, PointTracker, & PaleoGIS. Jan probably obtained a copy from me directly, or by using one of my programs. In either case, I deserve credit for the plate model (but not the paleogeography).
**Authors:** Sorry for the improper claim and citation. "are similar to those in Scotese (2004)" has been revised to "are based on Scotese (1997, 2004)" (line 100).

**Reviewer:** 106 "fossil collections" rather than "documented fossils"
**Authors:** We have modified this in the revision (line 116).

**Reviewer:** 116 This is an important sentence. It must be clear. Try, "The methodology can divided into three steps: 1) the original paleogeographic boundaries were restored to present-day coordinates by applying the inverse of the rotations used to make the reconstruction, 2) these restored boundaries were then rotated to new locations using the plate tectonic model of Matthews et al. (2016), finally, 3) the location of the paleocoastlines were adjusted using paleoenvironmental data from the Paleobiology database."

**Authors:** We have rewritten the sentence as suggested here (lines 126-130).

**Reviewer:** 117  Figure 2 illustrates the generalized workflow.
**Authors:** "a generalized workflow" has been revised to "the generalized workflow" (line 130).

**Reviewer:** 126  "to refine the rotations and ensure that the paleogeographic boundaries are restored accurately to their present-day locations."
**Authors:** We have modified the sentence in the revision (lines 138-140).

**Reviewer:** 141  Emphasize how tedious and labor intensive this procedure is. "The gaps and overlaps were fixed, feature by feature, map by map, by extending or modifying the outlines of each mismatched polygon in order to make the boundaries connect in a similar fashion to the original paleogeographies."
**Authors:** We have clarified this in the Discussions section in the revision (lines 346-348).

**Reviewer:** 151  Try "Once the gaps and overlaps were fixed, the reconstructed paleocoastlines were compared with the data from the PaleoBiology Database that described the marine and terrestrial environments of the fossil collections.  These comparisons were aimed at indentifying the differences between the mapped paleocoastlines and the marine and terrestrial environments in order to modify the location of the paleocoastlines."
**Authors:** We have revised this part in the revision as suggested here (lines 161-164).

**Reviewer:** 155  change "Only the fossils" to  "Only the fossil collections"
**Authors:** We have replaced "Only the fossils" by "Only the fossil collections" in the revision (line 166).

**Reviewer:** 157 change "fossils" to "collections"  and "Fossils" to "Fossil collections"
**Authors:** We have modified this throughout the manuscript.

**Reviewer:** 161-165   The sentence starting with "Alternatively . . " and everything after it, should be deleted. It is unnecessary.  Makes things unnecessarily complex.
**Authors:**  We have deleted this part in the revision.

**Reviewer:** 169  "collections were then attached"  - delete "motion"
**Authors:** We have deleted "motion" in the revision (line 175).

**Reviewer:** 170  Try, "Subsequently, a point-in-polygon test was used to determine whether the indicated terrestrial or marine fossil collection lied within the appropriate marine or terrestrial paleogeographic polygon. The results of these tests is discussed in the following section. (delete the rest of this paragraph).
**Authors:** We have modified this part in the revision as suggested (lines 176-178).

**Reviewer:** 177-178.  "In the next step, we modified the location of the paleocoastlines based on the differences between the paleoenvironments indicated by the fossil collections

and the mapped paleogeography.  Figures 4 & 5 illustrate how the paleocoastlines were modified. "

**Authors:** We have amended this part in the revision as suggested (lines 180-181).

**Reviewer:** 184  ". . . taken into account.  (3) The boundaries . . ."
**Authors:** We have deleted "as valid proxies to improve marine-terrestrial boundaries" in the revision (line 185).

**Reviewer:** 192 "to maximize  the use of  the paleoenvironmental information from the fossil collection  to improve . . "
**Authors:** We have changed "paleobiology" to "the paleoenvironmental information from the marine fossil collection" in the revision (lines 189-190).

**Reviewer:** 205 " when using the fossil collections. . "
**Authors:** We have replaced "paleobiology" by "the fossil collections" in the revision (lines 201-202).

**Reviewer:** 208 "deceptive fossils, however, are rare."
**Authors:** We have revised "deceptive fossils are rare." to "Such instances of deceptive fossil data are a potential limitation within our workflow, which we seek to minimise for example by excluding inconsistent fossils more than 500 km from previously interpreted paleoshorelines as described above." (lines 204-206)

**Reviewer:** 211  "4.1 Paleoenvironmental Tests"   - no Paleobiology used here.
**Authors:** We have modified "4.1 Paleobiology Tests" to "4.1 Paleo-evironmental tests" (line 209).

**Reviewer:** 210 -254  I still think this "consistency/inconsistency ratio " is somewhat dubious due to the changing location of the coastline (see previous discussion).  Maybe if it were couched in terms of a "match ratio" , or "mixing ratio" rather than an "inconsistency ratio". A high mixing ratio (mixing of marine and terrestrial data) would indicate a widely fluctuating coastline.  A low mixing ration would indicate relatively stable shorelines. Again, what should be flagged as anomalous are marine data points far removed inland from coastlines (>500 km) or terrestrial data points far removed, oceanward of coastlines.  It seems nearly pointless to flag contrary indications that lie adjacent to the coastline.
**Authors:** Given that the coastlines on the paleo-maps used in this study represent maximum transgression surfaces, and we only use marine fossil collections to improve the paleo-coastline locations and the paleogeographic geometries in the revision, this is not the case anymore. We have used the marine fossil collections less than 500 km from the nearest coastlines in the new tests and have flagged all inconsistent marine fossil collections far removed inland from the coastlines (>500 km) with red point symbology on each time-interval map (see a set of maps in Supplement materials).

**Reviewer:** 254 "scarce, the fossil collections were of limited . ."
**Authors:** We have revised "paleobiology data is" to "the fossil collections are" (line 239).

**Reviewer:** 261 "Methods"

**Authors:** We have revised "Method" to "Methods" (line 245).

**Reviewer:** 264-267  Rewrite this sentence.
**Authors:** We have rewritten the sentence in the revision (lines 249-251).

**Reviewer:** 281-287  Rewrite, simplify, clarify.   "380-285,81-58, and 37-2 Ma" should be "30-285 Ma, 81-58 Ma, and 37-2 Ma"
**Authors:** We have rewritten the sentence and modify "380-285, 81-58, and 37-2 Ma" to "30-285 Ma, 81-58 Ma, and 37-2 Ma" in the revision (lines 268-269).

**Reviewer:** 313  NO.  The sea level curves of Haq et al. 1987 & are not inferred from the flooding ratios. They have a completely separate derivation. I would delete this sentence.
**Authors:** As we have deleted the comparison between continental flooding curves and published sea level fluctuations, this sentence has been deleted accordingly.

**Reviewer:** 310 – 323  These values are in good agreement with the flooding curve I have independently produced.
**Authors:** We have deleted the comparison between continental flooding curves and published sea level fluctuations. Instead, we compared the flooded continental area curve generated from our amended paleogeography to previously published ones (see revised Fig. 9).

**Reviewer:** 326  A similar pattern of changing areas was published by Worsley et al (1984), Fig. 7.
**Authors:** We have deleted the whole comparison between emerged land area, total land area and the evolution of strontium isotopes of marine carbonates in the revision.

**Reviewer:** 335  "402 Ma to 2 Ma"
**Authors:** We have deleted the paragraph in the revision.

**Reviewer:** 343-345   I don't understand what you're trying to say here.  Don't you mean "emerged", not "submerged"?.
**Authors:** We have deleted the paragraph in the revision.

**Reviewer:** 368 "utility" rather than "flexibility"
**Authors:** We have deleted the paragraph in the revision.

**Reviewer:** 372 "variable" rather than "flexible"
**Authors:** We have replaced "flexible" by "variable" in the revision (line 374).

**Reviewer:** 375 "using paleoenvironmental data obtained from fossil collections"
**Authors:** We have changed "using paleobiology data" to "using paleo-environmental information indicated by the marine fossil collections from the PBDB." in the revision (lines 377-378).

**Reviewer:** 397 Please include an acknowledgement to my help with the editing.

**Authors:** We sincerely thank the reviewer for his constructive reviews and suggestions, that we have acknowledged (lines 408-409).

Comments about Tables
**Reviewer:** Table 1 Nearly all of the Sloss Sequence designations are incorrect. See Table 1 Revisions. Also the timescale for the maps is not the latest ICS timescale (2012). This means the ages may be off by as much as 4-6 million years.
**Authors:** We have corrected the table in the revision (see revised Table 1).

**Reviewer:** Table 2   - OK
**Authors:** We have modified Table 2 in the revision (see revised Table 2).

Comments about Figures
**Reviewer:** Fig 1 I would arrange with oldest on bottom to match the timescale on the left.
**Authors:** We think the current arrangement in Fig .1 from old time to young time could better match the geological time scale.

**Reviewer:** Fig 2 change "Reverse Engineer" to " Restore to Present-day", change  "Fix gaps" to "Fix gaps and overlaps"
**Authors:** We have changed "Reverse Engineer" to "Restore to Present-day" (see revised Fig. 2). We only fix the gaps.

**Reviewer:** Fig 3 Excellent Figure!
Fig 4    Nicely done, very clear.
Fig 5    Very clear – though I am not sue the changes are significant.
**Authors:** Thank you. The changes are significant and please see revised Figs 4, 5, 6 and a set of maps in Supplement materials.

**Reviewer:** Fig 6 I would change it to "Match Ratio".  Otherwise clear.
**Authors:** We have amended the explanation of "Consistency ratio" in the text to be clearer.

**Reviewer:** Fig 7 These area nice set of maps. Well done.  I think the revised coastlines are fine, however the continental margins seem cartoonish and extend far beyond the COB. The size and placement of the mountains through time are very inconsistent.
**Authors:** Thank you. The paleogeographic geometries in this study are all originally obtained from Golonka et al. (2006)'s paleo-maps and we use the paleo-environmental data of the marine fossil collections from the Paleobiology Database to improve the paleo-coastline locations and the paleogeographic geometries. Improving the continental margins or the size and placement of the mountains are beyond the scope of this study.

**Reviewer:** Fig 8 Clear.
**Authors:** Thank you.

**Reviewer:** Fig 9 Potentially misleading.  Both 9a & 9b should be separate diagrams because the y-axis values are different, and not equivalent.  See text comments for elaboration.

**Authors:** We have deleted Figure 9a and b. Instead, we have compared the flooded continental area generated from our amended paleogeography to previously published ones (see revised Fig. 9).

Comments about References Cited
In good shape, only a few things
**Reviewer:** 41   Blakey, 2008, is Blakey, 2003 in References
**Authors:** Blakey (2008) was accidentally missing and we have added it to the reference list (lines 426-427).

**Reviewer:** 95  Domeier and Torsvik, 2014 is missing, but there is a Domeier, 2016 that is not cited in the text.
**Authors:** We have added Domeier and Torsvik (2014) and have deleted Domeier (2016) in the References (lines 431).

**Reviewer:** 311 & 312  There is no Haq et al., 2012 in the References; Haq et al, 2008?
**Authors:** We have deleted the comparison between continental flooding curves and published sea level curves so they are not cited anymore.

Comments about Supplementary Materials
**Reviewer:** Good to have a copy of Golonka (2006) included.  It would have been nice to have the rotation model used by Golonka included as well.  The link to the Supplement of Golonka (2007) is no longer active.
**Authors:** We have attached a copy of Golonka (2006)'s digitised paleogeographic maps and the rotation model in Supplementary materials.

**Reviewer:** I compared some of Golonka's original maps to the updated paleogeographies. In some cases I was not able to see any of the modifications (see Figure 3).  It would be good to have a complete set of maps with the red and green symbols plotted as in Figures 4 & 5. That way we could see what was changed.
**Authors:** The paleo-coastlines are significantly different, except for a few time-interval maps where have few fossil data. We have included a set of maps to demonstrate that (see a set of maps in Supplement materials).

**Reviewer:** When I loaded the Paleobiology data points in Gplates, I could not distinguish the "marine" from the "terrestrial" data points. The only attributes that I could discern were "plateid" and "end and start" times. The marine data and the terrestrial data should be in separate files.
**Authors:** We have provided consistent and inconsistent marine fossil collection data in separate files (see Supplement materials) as only marine fossil data are used in the revision.
**Reviewer:** The authors attempt to produce a flexible, digital representation of Earth's plates through most of the Phanerozoic. This representation should allow testing paleogeographic features of the original dataset against other datasets, adopting different rotation models as used in the original dataset, among other things. The authors then use a comparison of their original distributions of land and sea to that implied by the distribution of fossil organisms, to get a more accurate picture of the distributions of land and sea through Earth's history. These 'improved' distributions are then used for various comparisons with eustatic sea level curves and measures for continental weathering. Although the attempt to build a flexible model of Earth's plate movements through time is fine and useful, most of the subsequent comparisons are, in my view, redundant, insufficiently interpreted and discussed. Also the methods section needs improvements. In the present state I can only recommend to reject the manuscript, and to encourage the authors to focus on the core of their work (the model), to improve the methods section, and revamp their 'testing' and their discussion.

**Authors:** We thank the reviewer for his/her constructive review that has guided our revision of the manuscript. We have amended the paleogeographic model, given more detail in the Methods section, and changed the tests carried out on the paleogeographies using paleobiology. We have deleted the comparison between continental flooding curves and published sea level fluctuations as there may be some circularity in this comparison, and the comparison between emerged land area, total land area and the evolution of strontium isotopes of marine carbonates. Instead, we have compared our flooded continental area curve to previously published ones (see revised Fig. 9). We have estimated the terrestrial and oceanic areal change due to filling gaps and modifying the coastline locations and the paleogeographic geometries over time (see Fig. 10 in the revision), tested the marine fossil collection dataset used in this study for fossil abundances over time with two different time scales (see Fig.11 in the revision), and discussed the limitations of the workflow we develop in this study.

**Reviewer:** Detailed comments by line number: 106-108, there is another important bias in the PBDB: the uneven entry of fossil data.

**Authors:** We agree and have added this to the sentence in the revision (line 118).

**Reviewer:** 116-117, repetition

**Authors:** We have rewritten this sentence in the revision (lines 126-130).

**Reviewer:** 145-147, I have the feeling that the authors are trying to explain here which environmental types have gone into the gaps and overlaps, but I failed to understand it.

**Authors:** We have deleted this sentence to avoid any confusion.

**Reviewer:** 155-159, here the authors sometimes talk about 'fossil collections' and sometimes about 'fossils', though my impression is that they always mean 'fossil collections' – please be consistent here and throughout the ms in general.

**Authors:** Yes, they all mean 'fossil collections'. This has been corrected throughout the manuscript.

**Reviewer:** 187-190, unclear how it was decided which 'fossils' (by which the authors presumably mean 'fossil collection site') are included in such a cluster and which aren't. It is important to make clear how the boundaries of these clusters are drawn.
**Authors:** In our revised version of the maps, we only use marine fossil collections to improve paleo-coastline locations and the paleogeographic geometries (see revised Figs 4, 5, 6), because the coastlines on the paleo-maps used in this study represent maximum transgression surfaces, so this is not the case anymore.

**Reviewer:** 235-243, this entire test is redundant: if you're adjusting the land-sea boundary in such a way that most inconsistencies are removed, of course does your 'consistency index' improve.
**Authors:** We have deleted the test of modified paleogeography with paleobiology, and only presented the test of unmodified paleogeography (see revised Fig. 6).

**Reviewer:** Paragraph 245-257, it is not clear to me what the authors are getting at with this paragraph. They discuss various biases and inhomogeneities of the fossil data, but neither do they apply a coherent test to the problem, nor do they reach any conclusion (except perhaps for "fewer fossils = fewer possibilities for adjustments", but this again is trivial).
**Authors:** We have carried out a test on the marine fossil collection dataset used in this study for fossil abundances over time with two different time scales: ICS2016 and Golonka (2000) (see revised Table 1), and we have revised this paragraph (lines 231-240), deleted the trivial part, presented the result (see Fig. 11 in the revision) and discussed it in the Discussions section (lines 325-339).

**Reviewer:** 245-249, as for lines 106-108, uneven entry of data is another potential bias.
**Authors:** We have added this in the revision (lines 235).

**Reviewer:** 249-251, "shorter time spans contain fewer fossils" – it might be interesting to systematically test the fossil dataset for this.
**Authors:** We have tested the dataset used in this study for fossil abundances over time with two different time scales: ICS2016 and Golonka (2000) (see revised Table 1), presented the result (see Fig. 11 in the revision) and discussed it in the Discussions section (lines 325-339).

**Reviewer:** 253, "biological organisms" – organisms are biological by definition
**Authors:** We have removed "biological" in the revision.

**Reviewer:** 264-267, here I was wondering how much of the "areal change" might relate to the gap filling and overlap removal that the authors have done to fit the plate reconstructions. In their lines 144-145 they wrote that the total areal variations ranged from 5.8 to -2.7%. A comparison of these values through time to the extent of area change through time (or something along these lines) might provide valuable insights here.
**Authors:** We have estimated the areal change in two key steps of the methodology, including filling gaps and modifying the coastline locations and paleogeographic geometries,

presented the results (see Fig. 10 in the revision) and explained it in the Discussions section (lines 303-323).

**Reviewer:** 281ff, unless I've overlooked it, there is a step missing here in the explanation of the method. So far, the authors explained that in their adjustments, they exchanged 'land' for 'sea' and vice versa. But now they start discussing the quantification of different habitat types (shallow vs. deep sea, mountains vs. low lands etc.). Does this mean that when the land-sea boundary was shifted, for example, the 'new sea area' was assigned the habitat type of the fossil collection that caused the change? For example, has an area previously classified as 'mountain' sometimes been replaced by 'shallow marine' and sometimes by 'deep marine'? If so, this needs to be explained in the Methods section.
**Authors:** We have explained this in the Methods section (lines 188-189).

**Reviewer:** 310ff, this whole paragraph seems redundant. It is pretty obvious to any earth scientist that continental flooding and eustatic sea level changes are linked. Not only is it obvious that eustatic sealevel changes cause continental flooding (what else should it be?); to make matters worse, the eustatic sealevel curves are inferred from the continental flooding history as recorded in the sedimentary record so you might be looking at circularity here.
**Authors:** We have removed this entire paragraph as indeed there could be some degree of circularity.

**Reviewer:** 332, the difference between 27.7% and 27.5% isn't really great, isn't it? The authors should be a little more cautious about the errors in their own model. Could this difference of 0.2% again result from their gap filling procedure? Or could it be related to the inconsistencies in their 'improved paleogeographies'? In their lines 238-241 they write that even their 'improved paleogeographies' are still 3-5% inconsistent, which is a lot more than the 0.2% difference mentioned above. I recommend that the authors assess these inherent errors in their model (gap filling and 'consistency' index) and then discuss only variations that exceed those errors.
**Authors:** Since we have deleted the comparison between emerged land area, total land area and the evolution of strontium isotopes, this part has been removed accordingly. As suggested here, we have amended the paleogeographic model and updated the test carried out on the paleogeographies using paleo-environments indicated by marine fossil collections from the PBDB. We have estimated the errors of two key steps in the workflow, including filling gaps and modifying the coastline locations and the paleogeographic geometries, on the terrestrial areal change over time (see Fig. 10 in the revision) and discussed them in the Discussions section (lines 303-323).

**Reviewer:** 341, 3% of the world's continental area has disappeared in the Neogene? Where did it go?
**Authors:** The Neogene increase in mountainous areas results in a net loss of continental area.

**Reviewer:** 350-351, the abbreviation CGM is not explained (and perhaps not necessary?)
**Authors:** As we have deleted this entire paragraph, this has been deleted in the revision accordingly.

**Reviewer:** 363, I find it dubious to 'confirm that Sr isotope ratios have a good correlation with emerged land areas' when there is no such correlation in the Paleozoic. Doesn't this rather indicate that there may be something fundamentally wrong with this correlation? I have no solution to the problem, but it seems more scientifically to me to point out such inconsistencies rather than to uncritically reiterate some lukewarm 'conventional wisdom'.
**Authors:** We have deleted the comparison between emerged land area, total land area and the evolution of strontium isotopes of marine carbonates in the revision.

**Reviewer:** 366ff, the 'Conclusions' nicely sum up the good parts and the problems of this study. The first paragraph outlines the good part, the flexible, digital plate model that could surely be of use for a wide range of earth scientists. The second paragraph discusses the redundant correlation between emerged land and eustatic sea level changes, and the third paragraph again 'confirms' a correlation between Sr isotopes and emerged land, which apparently doesn't exist in the Paleozoic.
**Authors:** Our conclusions have been amended in the revision, based on the input from the reviewer.

**Reviewer:** Table 1. why is this awkward Sloss 1988 timetable used? As far as I can tell, it applies to the US only, and connecting it to the accepted ICS and GSA timescales and to the periods, series and stages that have been used by geologists for more than 100 years is confusing. Avoid this, it is of no use for geologists and paleontologists.
**Authors:** Sloss (1988) is the base of the time scale of Golonka (2000) applied to the paleogeography used in this study. We have converted the time scales of Sloss (1988) and Golonka (2000) to agree with the ICS2016 and presented them together in the table (see revised Table 1).

**Reviewer:** Table 2, I had difficulties relating this table to what's written in the manuscript. The table distinguishes three paleogeographies (shallow marine, landmass/mountain, ice sheet), whereas in the text and fig 8 five distinctions are made (shallow marine, deep marine, land masses, mountains, ice sheets). Please be consistent here.
**Authors:** We have corrected this in the revision (see revised Table 2).

**Reviewer:** Figure 5. colors and shapes are not explained; perhaps refer to fig. 4? And I presume you mean "fossil collection sites" rather than "fossils"? I don't see any fossils in this figure.
**Authors:** We have replaced Figure 5 by a new figure (see Fig. 5 in the revision) in which the colours and shapes have been explained clearly. Yes, we refer to "fossil collection sites" rather than "fossils" and we have corrected this throughout the manuscript.

[revised manuscript text omitted]

---

## Author Response (AR3)

**Improving global paleogeography since the late Paleozoic using paleobiology**

Wenchao Cao*[,1], Sabin Zahirovic[1], Nicolas Flament[†,1], Simon Williams[1], Jan Golonka[2] and R. Dietmar Müller[1]

[1] EarthByte Group, School of Geosciences, The University of Sydney, NSW 2006, Australia

[2] Faculty of Geology, Geophysics and Environmental Protection, AGH University of Science and Technology, Mickiewicza 30, 30-059 Kraków, Poland

*Correspondence to: Wenchao Cao (wenchao.cao@sydney.edu.au)

[†]Current address: School of Earth and Environmental Sciences, University of Wollongong, Northfields Avenue, Wollongong, New South Wales 2522, Australia

This file contains the following content.

(1) Responses to comments from Referee #3 - Shanan Peters (Pages 2-5)

(2) Responses to comments from Anonymous Referee #4 (Pages 6-8)

(3) A marked-up version of the revised manuscript (Pages 9-33)

**Referee #3: Shanan Peters, peters@geology.wisc.edu**

**Reviewer:** Accurate paleogeographic reconstructions are required to test a wide range of hypotheses in the geosciences and they are a critical foundation upon which to build next-generation 4D Earth systems models. In order to produce the best possible reconstructions, no data can be left behind. This paper represents an important step towards a general method by which paleogeographically sensitive proxy data (e.g., fossils, sediments) can be used to test and improve existing paleogeographic reconstructions. This type of reconstruction model-data comparison is critical and the scale of the problem is such that algorithmic solutions and automation are of great added value. The paper is well written, the examples given are clear and powerful, and the potential of the approach and scientific outcomes are substantial. I have only a few suggestions, detailed below.

1) Discrete timescales of reconstructions and the mapping of proxy data therein.

Table 1 outlines the timescale used for the Golonka reconstructions and this represents the first major cull of Paleobiology Database data. Only those collections that are temporally resolved in the PBDB to fall entirely within the bins specified by the reconstruction timescale are included. An example bin used is the "late Cenomanian-early Campanian." According to the methods description, this means that any PBDB fossil collection assigned to a time interval of "Cenomanian" or "Campanian" would be omitted from the analysis and only fossil collections resolved to a finer biostratigraphic zone within these two international ages would be included. This is an understandable convention given the discrete bins of the original Golonka reconstructions, but this protocol results in a large cull of PBDB data. International ages (e.g., Cenomanian) are generally defined by our ability to consistently correlate globally. It is certainly the case that biostratigraphic zonation can be more precise, particularly in the marine record, but a stage-level age assignment for a given collection is something that PBDB data enterers would be very satisfied with in the majority of cases. Indeed, in the example of the Cenomanian and Campanian above, there are more than 2,200 collections resolved to these two intervals and these would not be included. Similar numbers of collections are probably omitted from all such divided international ages (this is somewhat conveyed in the differences between curves in FIg. 11). Does this cull matter? Given that the Cenomanian-Turonian sea level high stand is likely captured by at least some collections that are resolved only to "Cenomanian," its possible that it does. There are a few statistical approaches one could take to overcoming this problem (in addition to improving the PBDB ages constraints, see below).

Finally, it is noted that PBDB data were "downloaded" from the database on a given date. This is a useful description, but the PBDB API allows for very specific definition of download protocols in the form of a URL. I strongly recommend that these details either be specified or, better still, that the URL for the API request be provided (it might also be good to include citations that describe these PBDB resources). Requesting an official PBDB publication number, should the manuscript be accepted, and including that in the Acknowledgements would be appropriate as well. John Alroy and company's original vision and the many PBDB data contributors should be recognized in this capacity.

**Authors:** Yes, the abundance of fossil collections selected from the PBDB is sensitive to temporal resolution. Please refer to 5.3 "Marine fossil collection abundances in two different time scales" in the manuscript. In this study, we chose a conservative approach, only using fossil collections with temporal ranges lying entirely within the corresponding time intervals, to ensure that the resulting PBDB data do not unnecessarily "smear" the paleogeographic boundaries, although many fossil collections were ignored due to this selection criterion. One solution as the reviewer suggested here is improving the PBDB age constraints.

We specified the details of the downloaded information in the Supplementary Materials as the old PBDB API did not provide a URL for the paleobiology data when we requested them on 7 September 2016. We have acknowledged John Alroy et al.'s original vision and all the PBDB data contributors, and added an official PBDB publication number in the Acknowledgements.

**Reviewer:** 2) The "500 km test"

Fossil collections deviating by more than 500 km from previously interpreted paleoshorelines are excluded herein. Why 500km? Why not 397 km? Or 193 km? With few exceptions, whenever there is some value arbitrarily defined as a threshold there is a more interesting and principled approach. I would find the distribution of deviations between "previously interpreted" shorelines and PBDB collections fascinating, both in aggregate and on an interval-by-interval and/or a plate-by-plate basis. These distributions would have statistical utility of several different types, including defining a principled threshold criterion for data rejection. To me, this would be single most interesting analytical addition to the paper.

One potential utility of adopting an approach that leveraged the statistical nature of the distribution of deviations is that this could be used to add quantitative estimates of error (mostly of a random nature) that is introduced at every step, from the original coordinates of PBDB collections (some of which are most certainly not accurately located) to the assignment of an age.

**Authors:** We estimate the distances of marine fossil collections from the paleo-coastlines derived from the original paleogeographic maps of Golonka et al. (2006) since the Cretaceous period (see Figure 1 below). The result indicates that most marine fossil collections are within 500 km from the paleo-coastlines. We also have clarified this in the manuscript. In addition, we are trying to avoid the use of fossil data in cases where there may have been local/regional lakes or inland seas, that may not have been captured by the starting paleogeography. Please see the red points on Figure 4c in the manuscript and a set of PBDB test maps in the Supplementary Materials.

[Figure]

Figure 1. Cumulative frequency of the distance of the marine fossil collections from PBDB to paleo-coastlines derived from the paleogeographic maps of Golonka et al. (2006) since the Cretaceous period. Note that fossil collections located more than ~500 km away from paleo-coastlines represent outliers of their distribution.

**Reviewer:** 3) Minor points/questions: Use of fossils in pre-Triassic reconstructions, plates that appear during Phanerozoic, and next steps.

For geologically obvious reasons, not all plates can be tracked backwards in time to the Paleozoic. Are such plates excluded here? Does this matter to any of the results presented herein?

**Authors:** Yes, such plates and continental fragments are excluded. Excluding the plates that do not exist for the whole time period leads to discarding the fossil collections that are located on these plates.

**Reviewer:** Prior to the constraints on rotations provided by sea floor data, there is considerable uncertainty in the paleopositions of the continents, particularly with respect to longitude. At such times, the fossil record becomes increasingly important, not just for shoreline reconstructions but also for continent positions. I have previously detected a quantitative signature of PBDB biogeographic patterns in Paleozoic reconstructions and have concluded that at least some of the signal reflects the fact the the fossil record was relied on more heavily in the Paleozoic than in the post Paleozoic. Does this matter? Is a potentially changing role for fossils in deriving paleopositions detectable in any way here?

**Authors:** This point is interesting but in our view out of the scope of this study. This study mainly provides a workflow to revise the paleo-coastline locations and paleogeographic geometries using paleo-environmental information indicated by the marine fossil collections from the PBDB.

**Reviewer:** I'm excited by the possibility of completely closing the loop, from aggregation of paleogeographically-useful data in PBDB and Macrostrat-type resources to testing and improving paleogeographic reconstructions, in a largely automated, algorithmic fashion. Certainly there are things we are working on now to improve the temporal resolution of PBDB fossil collections, notably by integrating them with stratigraphic models such as those in Macrostrat. I'd be very interested in discussing how to improve this process and to fully capitalize on paleogeographic reconstructions in a way that avoids circularity problems and that makes the data as useful as possible to the GPlates team, and vice versa. This paper is an important and welcome first step.

**Authors:** We think it is a good idea to integrate stratigraphic data from Macrostrat Database to further constrain the paleogeographic reconstructions in a more automated and algorithmic fashion. We recognise the reviewer's ongoing work of improving the temporal resolution of PBDB fossil collections, which will greatly improve the availability of PBDB data. We welcome the reviewer's appreciation of our work.

**Anonymous Referee #4**

**Reviewer:** There is a growing appreciation of the wide and constructive uses of paleogeographic maps for many areas of the geosciences. The community is transitioning from "static" (as the authors call them) or non digital to digital format paleogeographic maps. As with many aspects of our science, the effort is somewhat scattered and in this case the University of Sydney group is a pioneer in this area.

The authors stated main contribution is to "develop a workflow to restore the ancient paleogeographic geometries back to their modern coordinates". There longer term goal is to provide the "first step towards the construction of paleogeographic maps with flexible spatial and temporal resolutions that are more easily testable and expandable with the incorporation of new paleo-environmental datasets…". These are worthy goals and I think this manuscript can make a contribution. But I found myself asking almost philosophical questions on the approach here rather than specific comments on the results. I think this manuscript is concise in its explanation of the methods and main results, but it needs expansion of the Discussion. My points below can mainly be addressed in an expansion of the 5.4 Limitations of the Workflow section into a more general Discussion section.

One major point I have is this. As the earlier reviewer #1 implies and I ask – should we use older paleogeographic reconstructions as is being done here, or essentially start over with more quantitative local to regional data and build up to global scale maps? Related to this - are the revised coastlines presented here an improvement over the Golonka 2006 coastlines, and how do we really know? A more general question is how to transition from the largely pre-digital era and publications with little (or at least much less) meta data and documentation of sources of data to the fully digital era? The lesser documentation of earlier paleogeographic research is mainly inevitable in that most (all?) older primary publications with proposed paleoenvironmental features (for example, coastlines) did not give quantitative spatial data in the modern sense. Note for example that the authors in their response to review #1 admit that: "Since the original data that were used to estimate the coastlines are not available for us, it is difficult to give the weight to the paleobiology data. The coastlines drawn on the original maps represent maximum transgression surfaces and we do not know much about their errors." This latter point is made by the authors despite the well-respected Professor Golonka as the lead author of the synthesis of the coastlines (his 2006 paper) used in the methods in the present paper. In summary, the current community that produces quantitative tectonic reconstructions and paleogeographic maps may have to admit that it might have to essentially "start over" with regional-scale studies in which paleobiological and paleoenvironmental data are entered on GIS-based or other spatially quantitative base maps rather than try to "improve" older maps.

**Authors:** Rebuilding a global scale paleogeographic maps requires access to the original data that were used to estimate paleogeographic maps. However, these original data are not available in the scientific community as they are confidential industrial data, even though Professor Golonka is a co-author of this study. The consistency ratio between the original paleogeographic maps of Golonka et al. (2006) and the paleo-environmental data from the PBDB used in this study is ~75% on average since the Devonian period. This indicates a good

quality and reliability of paleo-coastlines from Golonka et al. (2006). Ideally, we would build the paleogeographic maps from scratch, using a variety of data types with high spatial and temporal resolution. This is currently not possible, and is beyond the scope of our work.

As we note in the manuscript, the consistency ratio between the revised paleogeography and the paleo-environments indicated by the marine fossil collections is increased to nearly full consistency (100%). This indicates the revised coastlines are improved over the original coastlines of Golonka et al. (2006).

**Reviewer:** Another point of discussion – there are serious problems with the variability of the temporal resolution of the various paleoenvironmental data used here and this is only partly addressed. For example, maps attempting to show maximum transgressions imply that we know the coastline location at very specific times in the past (especially during Icehouse periods) with perhaps thousands to a few tens thousands years resolution. I agree with reviewer #1 that it might be best, and more appropriate to the data available, to locate only marginal marine zones or belts that suggest much wider temporal and spatial spans involved (10s to a few 100 thousands years). For example, individual fossil locations in PBDB have a temporal resolution dependent on the type of index fossil and where it lies in the paleontological record. Many index fossils have a resolution of many 100 thousands to 1-2 million years. My point is not that this variability of data resolution invalidates using these different data sets, but that there needs to be more discussion here of what exactly is being depicted on the paleogeographic maps and what a particular paleogeographic feature means in terms of temporal and spatial resolution.

**Authors:** Again, due to the inaccessibility of original data that were used to build the paleogeographic maps, we cannot estimate the temporal resolution of the coastline locations on the paleogeographic maps. A fossil location from the PBDB does have a temporal resolution. We have added some clarification in the 5.4 "Limitation of the workflow" section of the manuscript.

**Reviewer:** Finally, I think these points should be addressed with more discussion in the 5.4 Limitations of the Workflow section. How can the community best move to developing quantitative paleogeography maps? And what have the authors learned by trying to update older paleogeographic maps? The current discussion in section 5.4 of the PBDB is good and provides valid suggestions. Here is an example of the need for more discussion: you state in section 5.4 that "A remaining question is how to provide a continuous representation of paleogeographic change that combines continuous plate motion models with paleogeographic maps that do not explicitly capture changes at the same temporal resolution." I think the authors who just completed this workflow project should make suggestions on how to answer this question. I would also like the authors to directly address the earlier point I made on the value of revising older paleogeographic maps (Golonka or Blakey as examples) versus building new maps from more fragmentary, but spatially and temporally more quantitative, data from the PBDB, StratDB and Macrostrat.

**Authors:** From this study, we have learnt that (1) the paleogeographic maps of Golonka et al. (2006) are good estimates indicated by a high consistency with paleo-environmental data from the PBDB; (2) paleogeographically sensitive proxy data, such as paleo-environmental

data from the PBDB used in this study, can be used to test and improve existing paleogeographic reconstructions, which is the first step towards the construction of paleogeographic maps with flexible spatial and temporal resolutions.

In order to produce more quantitative paleogeographic maps, especially for local or regional paleogeographic reconstructions, it is important to integrate various data such as paleo-environment and paleo-lithofacies data, stratigraphic data from Macrostrat Database and StratDB Database, and tectonic settings.

[revised manuscript text omitted]

Legend items below the figure:

🔴 Inconsistent marine fossil collection    ⬛ Ice sheet    ⬜ Landmass    ⬜ Deep ocean

🟢 Consistent marine fossil collection    ⬛ Mountain    🟦 Shallow marine    🟦 Modified area

**Fig. 4. (a) Test between the global paleogeography at 76 Ma reconstructed using the plate motion model of**

605 **Matthews et al. (2016) with gaps fixed and the paleo-environments indicated by the marine fossil collections**

**from the PBDB. (b) Area modified (blue) to resolve the test inconsistencies. (c) Test between the revised**

**paleogeography at 76 Ma and the same marine fossil collections. Mollweide projection with 0°E central**

**meridian.**

[Figure]

Fig. 5. Test between unrevised and revised paleogeography at 76 Ma respectively and paleo-environments indicated by the marine fossil collections from the PBDB, and revision of the paleo-coastlines and paleogeographic geometries based on the test results, for southern North America (a, b, c), southern South America (d, e, f), northern Africa (g, h, i) and India (j, k, l). Regional Mollweide projection.

[Figure]

**Fig. 6. (a) Consistency ratios between global paleogeography with gap filled, but before PBDB test for the period 402-2 Ma, reconstructed using the plate motion model of Matthews et al. (2016) and the paleo-environments indicated by the marine fossil collections from the PBDB. (b) Numbers of consistent (light grey) and inconsistent (dark grey) marine fossil collections used in the tests for each time interval from 402 Ma and 2 Ma.**

[Figure]

396 Ma  368 Ma  348 Ma  328 Ma  302 Ma  287 Ma  277 Ma  255 Ma  232 Ma  218 Ma  195 Ma  169 Ma

Ice sheet     Mountain     Landmass     Shallow marine     Deep ocean

[Figure]

**Fig. 7. Global paleogeography from 402 Ma to 2 Ma reconstructed using the plate motion model of Matthews et al. (2016) and revised using paleo-environmental data from the PBDB. Black toothed lines indicate subduction zones, and other black lines denote mid-ocean ridges and transforms. Grey outlines delineate reconstructed present-day coastlines and terranes. Mollweide projection with 0°E central meridian.**

630

635

[Figure]

**Fig. 8. Global paleogeographic feature areas as percentages of Earth's total surface area estimated from the revised paleogeographic maps from 402 Ma to 2 Ma.**

[Figure]

640

**Fig. 9. Global flooded continental area since the Early Devonian Period from the original paleogeographic maps of Golonka et al. (2006) (grey solid line) and from the revised paleogeography in this study (pink line). Results for Blakey (2003, 2008), Golonka (2007b, 2009, 2012), Ronov (1994), Smith et al. (2004), Walker et al. (2002) are as in van der Meer et al. (2017). The van der Meer et al. (2017) curve (green line) is derived from**
645 **the strontium isotope record of marine carbonates.**

[Figure]

650

**Fig. 10. Terrestrial areal change due to filling gaps and modifying the paleo-coastlines and paleogeographic geometries over time. Green: based on the original paleogeographic maps of Golonka et al. (2006); Red: based on paleogeography reconstructed using a different plate motion model of Matthews et al. (2016) and gaps filled; Blue: based on paleogeography with gaps fixed and revised using the paleo-environments**

655 **indicated by marine fossil collections from the PBDB.**

[Figure]

660 **Fig. 11. Fossil abundance test on the marine fossil collection dataset used in this study with two different time scales: Golonka (2000) and ICS2016 (Table 1).**